# Fear of pain moderates the relationship between self-reported fatigue and methionine allele of catechol-O-methyltransferase gene in patients with fibromyalgia

**David Ferrera**[1]*, **Francisco Mercado**[1], **Irene Peláez**[1], **David Martínez-Iñigo**[1], **Roberto Fernandes-Magalhaes**[1], **Paloma Barjola**[1], **Carmen Écija**[1], **Gema Díaz-Gil**[2], **Francisco Gómez-Esquer**[2]

1 Department of Psychology, School of Health Sciences, Rey Juan Carlos University, Madrid, Spain,
2 Department of Basic Health Sciences, School of Health Sciences, Rey Juan Carlos University, Madrid, Spain

* david.ferrera@urjc.es

**Data Availability Statement:** The datasets used and/or analysed during the current study are

## Abstract

Previous research has shown a consistent association among genetic factors, psychological symptoms and pain associated with fibromyalgia. However, how these symptoms interact to moderate genetic factors in fibromyalgia has rarely been studied to date. The present research investigates whether psychological symptoms can moderate the effects of catechol-O-methyltransferase on pain and fatigue. A total of 108 women diagnosed with fibromyalgia and 77 healthy control participants took part in the study. Pain, fatigue, and psychological symptoms (anxiety, depression, pain catastrophizing, fear of pain and fear of movement) were measured by self-report questionnaires. Two types of statistical analyses were performed; the first was undertaken to explore the influences of *COMT* genotypes on clinical symptoms by comparing patients with fibromyalgia and healthy controls. In the second analysis, moderation analyses to explore the role of psychological symptoms as potential factors that moderate the relationship between pain/fatigue and *COMT* genotypes were performed. The main results indicated that patients carrying the Met/Met genotype reported significantly higher levels of fatigue than heterozygote carriers (i.e., Met/Val genotype) and higher levels of fatigue, but not significantly different, than Val homozygote carriers. Among patients with fibromyalgia carrying methionine alleles (i.e., Met/Met + Met/Val carriers), only those who scored high on medical fear of pain, experienced an intensified feeling of fatigue. Thus, the present research suggests that fear of pain, as a psychological symptom frequently described in fibromyalgia may act as a moderating factor in the relationship between the Met allele of the *COMT* gene and the increase or decrease in self-reported fatigue. Although further research with wider patient samples is needed to confirm the present findings, these results point out that the use of psychological interventions focused on affective symptomatology might be a useful tool to reduce the severity of fibromyalgia.

available in the OSF Home repository (DOI: 10. 17605/OSF.IO/M5QUK).

**Funding:** This work was supported by grants PSI2017-85241-R from the Ministerio de Economía y Competitividad (MINECO) of Spain, S2015/HUM-3327 EMO-CM, PEJD-2017-PRE/ SOC-3501 and SAPIENTIA-CMH2019/HUM-5705 from the Comunidad de Madrid.

**Competing interests:** The authors declare that the research was conducted in the absence of any commercial or financial relationships that could be construed as a potential conflict of interest.

## Introduction

Fibromyalgia is a multifactorial, but not yet fully understood, disease that is mainly characterized by chronic and diffuse musculoskeletal pain [1]. Multiple factors, such as sensory disturbances of pain perception [2, 3], dysfunctions of the central nervous system [4], stress [5] and environmental conditions [6] are very often involved in the development of fibromyalgia. Due to the complexity of fibromyalgia, other aetiological causes such as genetic factors, should not be excluded from the understanding of its development and maintenance [7]. Previous evidence related to familial aggregation studies has shown that having a first-degree relative affected by this disease increases the possibility of developing this syndrome 8.5 times [8–10].

The catechol-O-methyltransferase (*COMT*) gene has been repeatedly associated with the perception and experience of pain [11–13]. The enzyme related to this gene is functionally involved in the degradation of catecholamines [14, 15]. The *COMT* gene contains a single nucleotide polymorphism (SNP) resulting in a substitution of guanine for adenine at codon 158 (the Val158Met polymorphism). This change results in the amino acid-encoding gene being methionine (Met) instead of valine (Val) [11].

The Val158Met polymorphism has been associated with augmented central sensitization in patients with widespread chronic pain such as fibromyalgia [16, 17]. Some authors have indicated that patients carrying the Met/Met *COMT* genotype have greater sensitivity to painful stimuli [13, 18], along with a higher number of tender points [19]. The Met allele (i.e., Met/ Met + Met/Val genotype carriers) has been associated with higher levels of dopamine in the brain (primarily -but not exclusively- in the prefrontal cortex), which would lead to a decrease in the responsiveness of the endogenous opioid system through the regulation of enkephalins [20–22], favouring an increase in pain perception.

Other investigations involving healthy control and chronic pain samples grouped the Met/ Met and Met/Val genotypes (methionine carriers) into a single group to explore their influence on different symptoms compared to Val/Val genotype carriers [15, 19, 23–25]. This strategy was used because the Met allele (i.e., the Met/Met and Met/Val genotypes) is responsible for lower metabolic activity of the *COMT* enzyme, whereas the Val/Val genotype generates a fully active enzyme [26]. In this sense, it has been postulated that people carrying Met/Met or Met/Val genotypes are less able to tolerate pain than those carrying the Val/Val genotype due to a lower innate level of beta-endorphins [27].

In addition to pain, a persistent feeling of fatigue is usually reported as one of the most prevalent symptoms in this chronic disease [28, 29], so much so that patients perceive it as one of the most disrupting symptoms for daily living activities [30]. Fatigue can be experienced in fibromyalgia as a physical, emotional, or cognitive condition [31]. Due to its close linkage to pain perception (it has been observed that they share common physiological mechanisms), fatigue seems to contribute to the amplification of pain sensations [32]. On the other hand, fatigue has also been associated with discomfort, depression, anxiety, or catastrophizing thoughts, which often affect those with fibromyalgia [17, 33, 34].

Despite the extensive amount of scientific evidence about the symptomatology of fibromyalgia, fatigue has not been explored in relation to the *COMT* gene in this syndrome. However, some authors have found a relationship between the Met allele of *COMT* (i.e., lower *COMT* activity) and the risk of suffering from chronic fatigue syndrome [35, 36], a disease that shares some aetiological mechanisms with fibromyalgia [37]. Furthermore, other authors have reported that in breast cancer survivors, patients carrying the Met/Met genotype showed both higher levels of fatigue and pain perception [38].

Likewise, *COMT* polymorphisms, specifically the Val158Met polymorphism, have also been related to psychological symptomatology in fibromyalgia [12, 13, 23, 39, 40]. Depression,

anxiety, or pain catastrophizing, which have been associated with this syndrome, are also important contributors to pain intensification and discomfort [17, 33, 38]. Several investigations have shown a clear relationship between the Met/Met genotype and higher severity of depression and anxious symptomatology [13, 34]. Additionally, it has also been argued that the presence of high levels of pain catastrophizing, together with reduced *COMT* enzyme activity, could constitute a risk factor for the development of chronic pain [41]. Further investigations have confirmed the relationship between *COMT* polymorphism and pain catastrophizing in patients with fibromyalgia [40]. Indeed, these authors reported that patients who carry the Met/Met genotype reported more pain complaints when pain catastrophizing was augmented. The neurobiological mechanism by which affective symptoms may influence the relationship between the *COMT* gene and pain is still unclear. However, as a consequence of carrying the Met/Met genotype of the *COMT* gene, the influence of dopamine has been hypothesized to potentiate the amygdala response [42]. Thus, in healthy participants, carriers of the Met/Met genotype showed greater limbic activation when faced with negative stimuli than other genotype carriers [42, 43].

The lack of studies investigating the relationship between *COMT* genotypes, fatigue, and affective factors makes it necessary to explore the potential connections among them. Therefore, the aim of the present case-control study was to examine differences in pain and fatigue, as well as several other psychological outcomes, in patients with fibromyalgia associated with the *COMT* gene. More interestingly, we will explore how different affective and cognitive symptoms might moderate the relationship between specific genotypes of the *COMT* gene and self-reported fatigue and pain in patients with fibromyalgia. Based on past studies, we expected to find a relationship between the enzymatic activity of the Val158Met polymorphism and the severity of both self-reported pain and fatigue in fibromyalgia, in which affective and cognitive symptoms would act as moderating factors. Thus, patients showing the lowest *COMT* activity (i.e., Met allele carriers) would experience greater pain and fatigue severity as a function of the worsening of their psychological symptomatology.

## Materials and methods

### Participants

One hundred and eight women diagnosed with fibromyalgia and seventy-seven healthy control (HC) participants took part in the study. The patients fulfilled the diagnostic criteria for fibromyalgia established by the American College of Rheumatology (ACR) in 2010 [1]. Different rheumatologists from the Comunidad de Madrid (Spain) carried out the diagnosis of the fibromyalgia. Patients were recruited from the Fibromyalgia and Chronic Fatigue Syndrome Association (AFINSYFACRO) and the Fibromyalgia Association of Pinto (AFAP). HC participants were recruited by means of an emailed advertisement and public advertisements located within Rey Juan Carlos University.

Most of the patients with fibromyalgia took analgesics or anti-inflammatories at the time of the experiment. Patients taking other medications (benzodiazepines and antidepressants) continued to do so because they were essential medical prescriptions. The HC participants did not suffer any chronic pain conditions. The participants in both groups were required to be women of European descent. Neurological diseases or disorders that impair cognitive functions, psychosis, and substance abuse/dependence were the exclusion criteria. This study was approved by the Rey Juan Carlos University Research Ethics Board and followed all ethical requirements of the Helsinki Declaration. Written informed consent was obtained from all participants.

## Procedure

The data from the entire sample were collected between 2011 and 2015 in the School of Health Science of Rey Juan Carlos University (Madrid, Spain). Sociodemographic characteristics, along with data about medication consumption, medical/psychological history, health habits and saliva samples were collected for all participants. Once in the laboratory, participants also completed a visual analogue scale (VAS) for pain and fatigue in the previous week ranging from 0 (no pain/fatigue at all) to 10 (worse imaginable pain/fatigue). Subsequently, some self-report questionnaires were administered to participants for the assessment of psychological variables. Patients with fibromyalgia completed the Fibromyalgia Impact Questionnaire (FIQ) [44], a specific questionnaire that assesses their current health and functional status. Subsequently, HC participants and individuals with fibromyalgia completed the Beck Depression Inventory (BDI) [45] and the State-Trait Anxiety Inventory (STAI) [46], to measure depressive and state-trait anxiety symptoms, respectively. The Pain Catastrophizing Scale (PCS) [47] was also completed. The PCS assesses catastrophic thoughts in pain-related situations. The Tampa Scale for Kinesiophobia (TSK) was also used [48]. It is an instrument for measuring fear of movement or injury during everyday life activities. Finally, participants completed the Fear of Pain Questionnaire (FPQ-III) [49]. This scale consists of three subscales: severe pain, minor pain, and medical/clinical pain. The completion of these forms took approximately 30 to 45 minutes.

## COMT genotyping

Unstimulated whole saliva samples were collected into collection tubes according to standardized procedures. Participants were asked not to eat, drink or chew gum for 1 hour before the sample collection. Immediately after collection, the samples were stored at -20˚C until analysis.

Genomic DNA was extracted from 5 ml of saliva using a REALPURE Saliva RBMEG06 Kit (Durviz, Valencia, Spain) according to the manufacturer's protocol.

The resulting DNA was diluted to 100–1000 ng/μl, using 1×Tris-EDTA (TE) buffer (Sigma-Aldrich, Dorset, UK) and assessed for purity and concentration using a NanoDrop™ ND1000 Spectrophotometer (Thermo Fisher Scientific Inc., Hemel Hempstead, Hertfordshire, UK). *COMT* polymorphisms were genotyped by real-time polymerase chain reaction (RT-PCR) analysis using TaqMan® Predesigned SNP Genotyping Assays for rs4680 polymorphisms (Thermo Fisher Scientific Inc, Hemel Hempstead, Hertfordshire, UK). TaqMan® SNP Genotyping Assays use TaqMan® 5'-nuclease chemistry for amplifying and detecting specific polymorphisms in purified genomic DNA samples. Each assay allows the genotyping of individuals for a single nucleotide polymorphism (SNP). Each TaqMan® SNP Genotyping Assay contains: (A) sequence-specific forward and reverse primers to amplify the polymorphic sequence of interest and (B) two TaqMan® minor groove binder (MGB) probes with nonfluorescent quenchers (NFQ): one VIC™-labelled probe to detect Allele 1 sequence and one FAM™-labelled probe to detect Allele 2 sequence. Amplification was carried out using an ABI Prism 7000 Sequence Detection System (Thermo Fisher Scientific Inc., Hemel Hempstead, Hertfordshire, UK) in the Genomics Unit of the Technological Support Center (CAT) of Rey Juan Carlos University. All genotypes were determined in duplicate.

## Statistical analyses

The chi-square ($\chi^2$) test was used to assess the distribution of the genotypes between patients and HC participants to check for Hardy-Weinberg equilibrium (HWE). Hardy-Weinberg equilibrium determines which frequencies should be observed in the population for each

genotype based on the frequencies of the alleles. Under normal conditions, whether the transmission of alleles from parents to offspring is independent, the probability of observing a specific combination of alleles depends on the product of the probabilities of each allele. Before carrying out an association study, it must be verified if the balance is met in the control sample. This can also be done with the patients' sample and it is possible that the equilibrium does not meet the criteria. This may be indicate that this gene can be associated with the development of the given disease [50].

Subsequently, the effects of the *COMT* genotypes on fibromyalgia symptomatology were tested in two steps. First, we examined the possible relationship between the different *COMT* genotypes and the most characteristic clinical symptoms of fibromyalgia compared to HC participants. To this end, two-way ANOVAs were performed, in which the scores on the self-reported questionnaires were considered dependent variables and *COMT* genotypes (three levels: Met/Met, Val/Val and Met/Val) and the group (two levels: patients with fibromyalgia and HC participants) were considered factors. In all cases, post-hoc comparisons to determine the significance of the pairwise contrasts were performed using the Bonferroni procedure.

As mentioned, pain and fatigue are highly related, and this association was explored in two ways. First, at an exploratory level, Pearson correlations (r) were determined. Second, to test whether the effects on self-reported fatigue were strictly due to the *COMT* genotypes but not influenced by the self-reported pain scores, a series of ANCOVAs were performed with the aim of neutralizing its possible effect on the statistical results. Effect sizes were computed using the partial eta-square ($\eta^2_p$) method.

The second step aimed to explore the possible moderating effects of psychological symptoms (depression, anxiety, fear of movement, pain catastrophizing and fear of pain) on the relationship between *COMT* genotypes and self-reported fatigue and pain perception in patients with fibromyalgia. For this purpose, we carried out a series of multiple regression analyses. Criteria were regressed onto predictors, and then the interaction term (genotype x BDI, genotype x STAI, genotype x TSK, genotype x FPQ-III, genotype x PCS) was entered into the equation to test the regression coefficient significance.

Finally, the possible effect of medication on clinical symptoms (i.e., pain, fatigue, depression, anxiety, fear of pain, fear of movement and pain catastrophizing) within the fibromyalgia group was tested using one-way ANOVA, including patients using and not using particular medications using a previously reported method [51, 52]. These control analyses were conducted including analgesics, benzodiazepines and antidepressants, as factors. Effect sizes were computed using the partial eta-square ($\eta^2_p$) method. We also computed post-hoc statistical power in two ways. Whereas the observed power for the ANOVAs was computed using SPSS, the post-hoc power of the moderation analyses was estimated with G*Power [53]. A significance level of 0.05 (two-tailed) was used for all statistical analyses. All statistical analyses were performed with the SPSS package (v.22.0; SPSS Inc., Chicago; IL, USA).

## Results

### Genotype frequencies of the *COMT* gene

As previously explained, the chi-square ($\chi^2$) test was used to assess the distribution of the genotypes between patients and HC participants to check for Hardy-Weinberg equilibrium (HWE). In Table 1, statistical data related to genotypes and allele frequency distributions of the *COMT* gene considering each group of participants can be observed. The frequency of the Val158Met polymorphism distribution fulfilled the HWE for the HC participants ($\chi^2 = 1.065$; $p = 0.301$) and the patients ($\chi^2 = 0.040$; $p = 0.840$).

**Table 1. Allele and genotype frequencies of *COMT* in the patients with fibromyalgia and the healthy controls participants.**

| Genotypes | Genotype frequencies n (%) | | | | Allele frequencies | |
|---|---|---|---|---|---|---|
| | HC (n = 77) | Fibromyalgia (n = 108) | p-value | | HC (n = 77) | Fibromyalgia (n = 108) |
| Val/Val | 16 (20.8) | 25 (23.1) | 0.703 | Val | 0.487 | 0.486 |
| Met/Val | 43 (55.8) | 55 (50.9) | 0.509 | Met | 0.513 | 0.514 |
| Met/Met | 18 (23.4) | 28 (25.9) | 0.689 | | | |

## Demographic, clinical and psychological analyses

All participants were aged between 35 and 67 years old. Although significant differences were found between the groups for age [$F_{(1,183)}$ = 7.01; p = 0.008], we did not find significant differences in age among the three different *COMT* genotypes for fibromyalgia patients [$F_{(2,105)}$ = 0.254, p = 0.776] or for the HC group [$F_{(2,74)}$ = 1.982, p = 0.158].

On the other hand, ANOVAs showed a main effect of group, where fibromyalgia participants showed significantly higher scores than HC for the majority of the psychological symptoms, such as depression, state anxiety, trait anxiety and fear of movement. Additionally, the total PCS and its three subscales (PCS rumination, PCS magnification and PCS helplessness) also reached a higher degree of severity in patients with fibromyalgia. As expected, pain perception and fatigue perception (measured by VAS) were also significantly higher in individuals with fibromyalgia than in HC participants. There were no significant differences in total fear of pain or in any of their subscales (severe, minor or medical). Additionally, the consumption of drugs by patients was higher than that of participants belonging to the HC group. A summary of these statistical data can be seen in Table 2.

**Table 2. Mean and standard deviations of age, level of anxiety, depression, pain catastrophizing, fear of pain, fear of movement, pain and fatigue.** Information about the percentage of participants (healthy controls and patients) who were taking medications is also included.

| Variables | HC | Fibromyalgia | p-value |
|---|---|---|---|
| Age | 48.15 (9.39) | 51.86 (9.23) | **0.008** |
| Medication | | | |
| Antidepressant (%) | 0 | 33.3 | **0.0001** |
| Benzodiazepines (%) | 1.3 | 33.3 | **0.0001** |
| Analgesics (%) | 2.6 | 61.1 | **0.0001** |
| Others (%) | 16.9 | 46.3 | **0.0001** |
| STAI State | 24.12 (21.23) | 53.76 (26.52) | **0.0001** |
| STAI Trait | 30.45 (25.46) | 68.56 (28.27) | **0.0001** |
| BDI | 6.79 (7.33) | 26.38 (15.46) | **0.0001** |
| PCS Total | 29.43 (25.79) | 51.12 (27.55) | **0.0001** |
| PCS Rumination | 32.89 (28.89) | 44.69 (28.11) | **0.001** |
| PCS Magnification | 39.45 (24.29) | 55.00 (27.39) | **0.0001** |
| PCS Helplessness | 31.26 (25.12) | 54.98 (27.50) | **0.0001** |
| FPQ-III Total | 73.53 (21.99) | 74.17 (21.10) | 0.859 |
| FPQ-III Severe | 30.96 (10.63) | 32.08 (9.88) | 0.511 |
| FPQ-III Minor | 19.89 (8.19) | 19.61 (6.35) | 0.814 |
| FPQ-III Medical | 22.63 (6.40) | 22.42 (7.36) | 0.856 |
| TSK | 27.27 (12.02) | 40.87 (7.66) | **0.0001** |
| VAS Pain | 0.55 (1.17) | 6.18 (2.38) | **0.0001** |
| VAS Fatigue | 0.94 (1.52) | 5.33 (2.73) | **0.0001** |
| FIQ | - | 55.93 (22.58) | - |

## Effect of *COMT* genotypes and the group of the participants on the clinical variables

Additionally, ANOVAs conducted to investigate possible interaction effects between group and *COMT* genotypes (i.e., Met/Met, Met/Val and Val/Val) yielded statistical significance for fatigue [$F_{(2,152)}$ = 5.975; p = 0.003, $\eta^2_p$ = 0.073]. The observed power for the interaction effect between the *COMT* and group for fatigue yielded a high statistical power (1- β = 0.87). Post-hoc analyses showed differences only within the fibromyalgia group. Patients carrying both the Met/Met (M = 6.56, SD = 2.37) and the Val/Val (M = 5.99, SD = 2.62) genotypes reported significantly higher levels of fatigue than the subgroup carrying the heterozygous genotype (Met/Val genotype: M = 4.4, SD = 2.65) (see Fig 1), but unexpectedly, no significant differences were found in the post-hoc comparison between Met/Met and Val/Val carriers (p = 0.99). Although a trend was identified for the interaction between Group and *COMT* genotypes for pain perception, it did not reach statistical significance [$F_{(2,175)}$ = 2.504; p = 0.085, $\eta^2_p$ = 0.028]. Unlike in other studies, no significant differences were found between the Met/Met and Val/Val genotypes of the *COMT* gene in fibromyalgia. No other significant differences were found for the rest of the *COMT* and group interactions (see Table 3 for the full statistical details). Finally, in these analyses, no main effects of the *COMT* genotypes on any of the clinical variables were observed across groups (S1 Table).

As explained in the Statistical Analyses section, we explored the relationship between self-reported pain and fatigue in terms of Pearson correlations. Self-reported pain was positively correlated with self-reported fatigue (r = 0.781, p = 0.001). Therefore, the higher the score of self-reported fatigue was, the higher the score of self-reported pain. Subsequently, a series of ANCOVAs were carried out including self-reported pain as a covariable. The analyses revealed that the differences shown in the self-reported fatigue for the interaction between the *COMT* genotypes and group were independent of self-reported pain. In other words, the effect of *COMT* genotype and the group interaction on self-reported fatigue remained significant after controlling for the score on self-reported pain ($F_{(2,151)}$ = 3.590, p = 0.030, $\eta^2_p$ = 0.045).

To test whether the effects were strictly due to genotypes but not influenced by the difference in age between the groups, a series of ANCOVAs were carried out. The possible influence

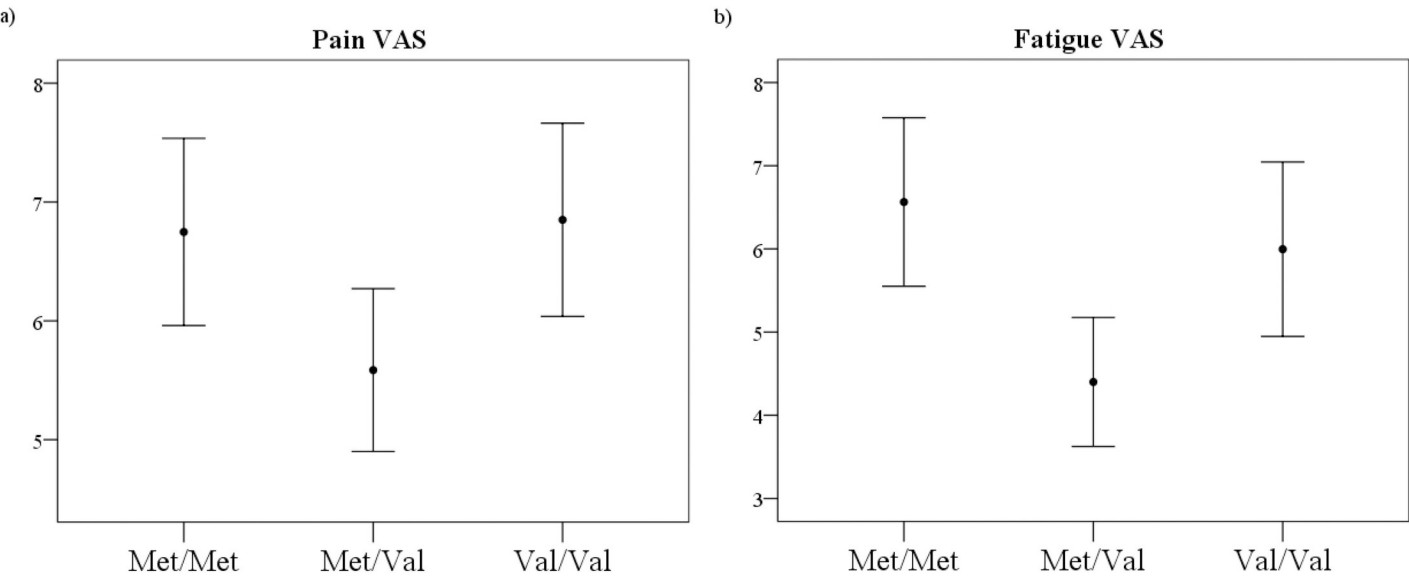

**Fig 1.** Mean and standard error of VAS pain (A) and VAS fatigue (B) for patients with fibromyalgia.

**Table 3. Mean scores and standard deviations (in parenthesis) of clinical measures.** Data are separated by group (fibromyalgia and healthy control) and genotypes (Val/Val, Met/Val and Met/Met). P-values for the interaction effects between *COMT* x Group were also included.

| | *COMT* | | | | | | p-value |
|---|---|---|---|---|---|---|---|
| | **Fibromyalgia** | | | **Healthy Control** | | | |
| | **Met/Met** | **Met/Val** | **Val/Val** | **Met/Met** | **Met/Val** | **Val/Val** | |
| STAI State | 50.84 (27.24) | 54.54 (27.42) | 54.85 (24.77) | 29.94 (26.01) | 21.07 (18.30) | 25.81 (22.56) | 0.386 |
| STAI Trait | 74.28 (20.90) | 64.96 (31.35) | 70.53 (27.55) | 33.94 (26.68) | 29.34 (25.06) | 29.50 (26.47) | 0.824 |
| BDI | 25.44 (15.41) | 25.05 (14.49) | 29.85 (17.29) | 9.11 (10.48) | 6.41 (6.25) | 5.18 (5.35) | 0.299 |
| PCS Total | 59.45 (23.08) | 47.02 (28.83) | 51.62 (27.90) | 31.37 (23.33) | 29.34 (26.54) | 27.30 (28.41) | 0.591 |
| PCS Rumination | 48.31 (29.11) | 42.95 (28.01) | 44.81 (28.27) | 32.87 (22.27) | 33.51 (30.72) | 31.23 (31.24) | 0.851 |
| PCS Magnification | 60.50 (25.03) | 48.33 (29.24) | 62.37 (23.53) | 43.25 (20.85) | 39.05 (24.54) | 35.84 (28.55) | 0.252 |
| PCS Helplessness | 67.31 (23.66) | 49.83 (27.23) | 54.11 (28.65) | 33.93 (24.44) | 30.40 (25.72) | 30.30 (26.07) | 0.413 |
| FPQ-III Total | 79.09 (22.73) | 69.08 (21.69) | 79.38 (16.66) | 78.43 (22.71) | 73.18 (21.47) | 66.80 (22.83) | 0.201 |
| FPQ-III Severe | 33.66 (10.07) | 30.38 (10.85) | 33.88 (7.36) | 30.68 (10.57) | 31.53 (10.78) | 29.60 (11.22) | 0.383 |
| FPQ-III Minor | 21.61 (7.29) | 17.87 (5.96) | 21.15 (5.54) | 22.06 (9.58) | 19.75 (7.70) | 16.90 (7.01) | 0.136 |
| FPQ-III Medical | 23.80 (7.88) | 20.74 (7.17) | 24.34 (6.81) | 25.50 (6.97) | 21.93 (5.79) | 20.30 (6.39) | 0.171 |
| TSK | 40.73 (6.86) | 41.09 (7.85) | 40.61 (8.18) | 30.8 (8.57) | 28.20 (12.40) | 20.88 (13.03) | 0.137 |
| VAS Pain | 6.74 (1.96) | 5.58 (2.54) | 6.85 (2.15) | 0.70 (1.35) | 0.58 (1.16) | 0.33 (1.01) | 0.085 |
| VAS Fatigue | 6.56 (2.37) | 4.40 (2.65) | 5.99 (2.62) | 0.55 (1.03) | 1.20 (1.73) | 0.77 (1.46) | **0.003** |

of age was neutralized by introducing it as a covariable. ANCOVAs revealed that the influence of the *COMT* genotypes on clinical outcomes was independent of age (for both patients with fibromyalgia and HC participants). Specifically, the interaction effect between group and the *COMT* gene for fatigue remained significant [$F_{(2,151)} = 6.44$; $p = 0.002$, $\eta^2_p = 0.079$], as well as maintaining a tendency towards significance for pain [$F_{(2,174)} = 2.629$; $p = 0.075$, $\eta^2_p = 0.029$]. No other significant difference was found for the rest of the statistical contrasts.

## Moderation analyses

To explore the potential moderating role played by psychological symptomatology on the relationship between the three genotypes of the *COMT* Val158Met polymorphism and pain or fatigue, moderated multiple regression analyses were carried out. Whereas the *COMT* genotype was introduced as an independent variable, psychological scores were set as moderated variables. Self-reported pain and fatigue were considered dependent variables. The results did not show moderation effects for any of the tested variables, either for self-reported pain or for self-reported fatigue ($p > 0.05$). Subsequently, another complementary moderation analysis was carried out, grouping the methionine carriers (i.e., Met/Met + Met/Val) into a single group and comparing them with the homozygous valine participants (i.e., Val/Val carriers), as has been repeatedly done in several previous studies [15, 19, 23–25]. A significant interaction effect was found between the *COMT* genotypes and medical fear of pain for self-reported fatigue ($\beta = -0.250$; $p = 0.031$). The analyses performed by using G*Power for the effects of moderation analyses between *COMT* and self-reported fatigue moderated by the level of fear of pain showed a high statistical power ($1- \beta = 0.99$). Additionally, the fatigue regression equation was estimated for each group and showed that patients carrying Met alleles and scoring higher on medical fear of pain reported enhanced levels of fatigue ($\beta = 0.4$; $p = 0.001$), as was hypothesized (see Table 4). However, for participants carrying Val/Val, the medical fear of pain did not predict differences in fatigue ($\beta = - 0.134$; $p = 0.531$) for this group (see Fig 2).

Finally, the analysis conducted to examine potential psychological moderators in the relationship between the *COMT* gene and pain perception also suggested a significant

**Table 4. Results of the multiple regression analysis for the interaction term (*COMT* x FPQ-III medical) for fatigue and pain equation in individuals with fibromyalgia.**

| Dependent variables | Step | Predictors | Standardized Coefficients | t | R² | ΔR² | p |
|---|---|---|---|---|---|---|---|
| Fatigue | | | | | | | |
| | 1 | *COMT* | 0.103 | 1.007 | 0.094 | 0.094 | 0.317 |
| | | FPQ-III medical | 0.276 | 2.697 | | | 0.008 |
| | 2 | *COMT* | 0.130 | 1.288 | 0.142 | 0.048 | 0.201 |
| | | FPQ-III medical | 0.392 | 3.462 | | | 0.001 |
| | | *COMT* x FPQ-III medical | -0.250 | -2.199 | | | 0.031 |
| Pain | | | | | | | |
| | 1 | *COMT* | 0.107 | 1.032 | 0.046 | 0.046 | 0.305 |
| | | FPQ-III medical | 0.168 | 1.622 | | | 0.108 |
| | 2 | *COMT* | 0.143 | 1.378 | 0.083 | 0.037 | 0.172 |
| | | FPQ-III medical | 0.277 | 2.367 | | | 0.020 |
| | | *COMT* x FPQ-III medical | -0.227 | -1.911 | | | 0.059 |

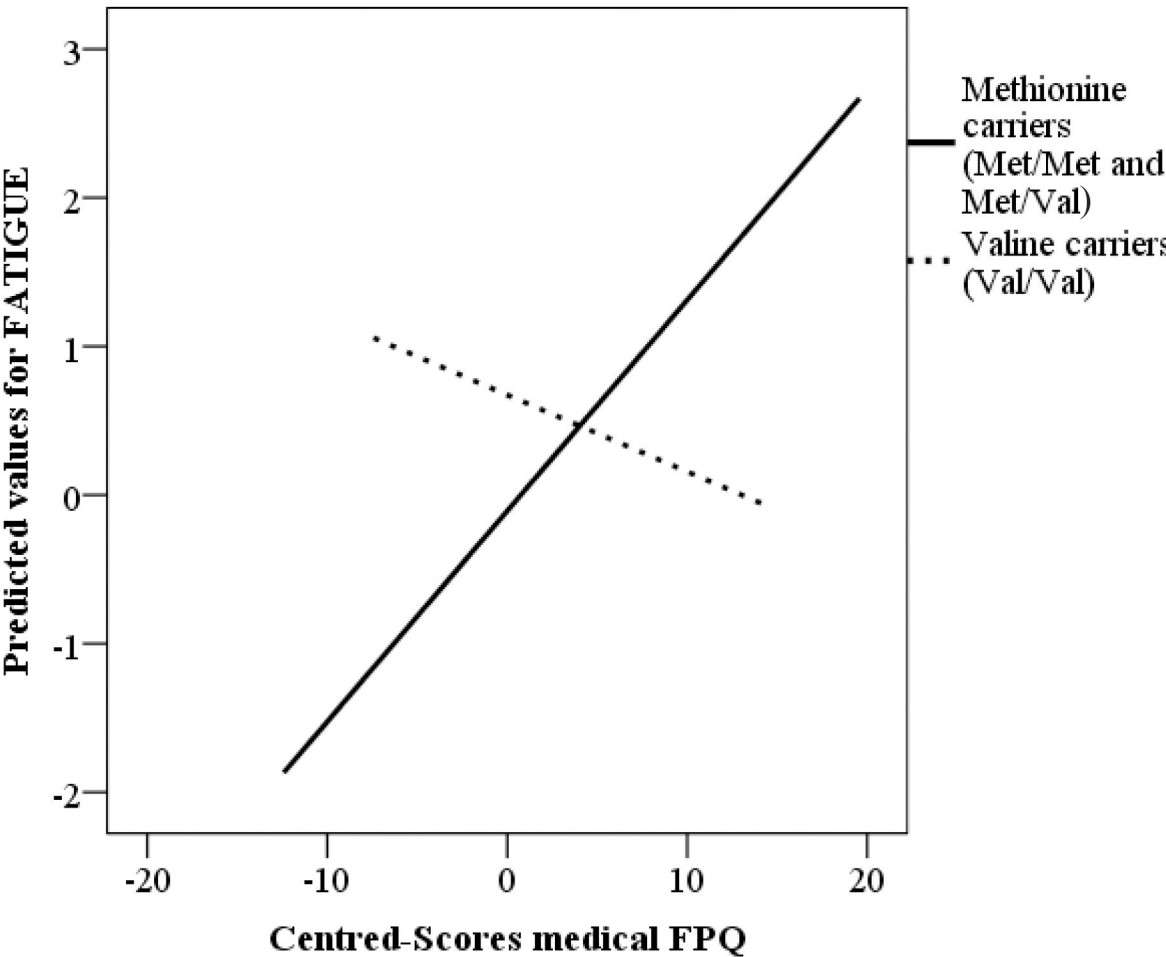

**Fig 2. Medical fear of pain moderation and its relationship of *COMT* and fatigue.**

**Table 5. P-values related to the medication effect (analgesics, benzodiazepines and antidepressants) on clinical measures (pain, fatigue, anxiety, depression, pain catastrophizing, impact of fibromyalgia, fear of pain and fear of movement) within the patients´ group.**

| Clinical variables | Medication | | |
|---|---|---|---|
| | Analgesics | Benzodiazepines | Antidepressants |
| VAS Pain | 0.220 | 0.619 | 0.672 |
| VAS Fatigue | 0.152 | 0.288 | 0.450 |
| STAI State | 0.547 | 0.323 | 0.282 |
| STAI Trait | 0.190 | 0.889 | 0.373 |
| BDI | 0.803 | 0.137 | 0.179 |
| PCS Total | 0.875 | 0.907 | 0.953 |
| PCS Rumination | 0.579 | 0.700 | 0.919 |
| PCS Magnification | 0.658 | 0.485 | 0.974 |
| PCS Helplessness | 0.652 | 0.485 | 0.711 |
| FPQ-III Total | 0.878 | 0.720 | 0.617 |
| FPQ-III Severe | 0.928 | 0.546 | 0.743 |
| FPQ-III Minor | 0.481 | 0.469 | 0.484 |
| FPQ-III Medical | 0.847 | 0.288 | 0.963 |
| TSK | 0.920 | 0.563 | 0.327 |
| FIQ | 0.086 | 0.181 | 0.175 |

involvement of medical fear of pain ($\beta$ = -0.227; p = 0.059) in such a way that Met carriers showing higher levels of medical fear of pain experienced a more intense pain perception ($\beta$ = 0.271; p = 0.025). However, in those participants carrying Val/Val, the medical fear of pain did not predict differences in pain perception ($\beta$ = -0.193; p = 0.344). In this case, the effects of moderation analyses between *COMT* and pain moderated by levels of fear of pain also showed high statistical power (1- $\beta$ = 0.99). For the rest of the interactions, the regression coefficients were not significant (see Table 4).

## Potential effects of medication on the clinical symptoms of patients with fibromyalgia

Table 5 summarizes clinical data from the patients according to the different types of medication taken (i.e., analgesics, benzodiazepines, and antidepressants). ANOVAs did not reveal any significant effect on the clinical symptomatology related to the intake of medications (all *p-values* were above 0.1), indicating that the severity of pain, fatigue or psychological symptoms was similar between the patients with fibromyalgia who were taking medications and those who were not.

## Discussion

The present results indicate that different genotypes of the Val158Met *COMT* polymorphism might contribute to the individual differences related to fatigue symptoms in fibromyalgia syndrome. To the best of our knowledge, our results are the first to show that patients carrying the Met/Met genotype (i.e., low *COMT* activity) exhibited significantly higher levels of self-reported fatigue than heterozygote carriers. In addition, Met/Met genotype carriers also reported higher levels of self-reported fatigue than valine homozygotes, but unexpectedly, these differences were not significant. More interesting is the finding showing the role of some psychological symptoms (i.e., medical fear of pain) in moderating the relationship between *COMT* genotypes and self-reported fatigue. Indeed, patients with fibromyalgia carrying the methionine allele and with higher scores of medical fear of pain reported a more intense

feeling of fatigue than patients carrying the Val/Val genotype of the *COMT* gene. The novelty of these results deserves careful interpretation.

As mentioned, our results indicated that patients carrying the Met/Met genotype of the *COMT* gene showed significantly more self-reported fatigue than those carrying the Met/Val genotype, but not significant higher, than the valine homozygote carriers. The lack of available empirical evidence linking self-reported fatigue and *COMT* genotypes in fibromyalgia makes it difficult to provide a straightforward explanation of these findings. Nonetheless, the present data are in alignment with several previous studies showing a relationship between *COMT* and self-reported fatigue in other pathologies (e.g., breast cancer) and healthy participants [38, 54–56], supporting the idea that the *COMT* gene may influence fatigue perception. More precisely, Met/Met genotype carriers reported an increase in fatigue levels [38]. This fact may have various complementary and non-exclusive interpretations. Some authors have explained that a reduction in *COMT* enzymatic activity (i.e., the Met/Met genotype) entails a reduction in the density of endogenous opioid receptors [20], which, in turn, increases pain sensitivity. This mechanism could explain the enhancement of fatigue perception as a consequence of the common pathophysiological mechanisms shared between pain and fatigue [32]. On the other hand, animal models have shown that low *COMT* activity might lead to an increase in cytokines [57]. This fact could contribute to the increased fatigue perception in Met/Met genotype carriers since cytokines could cause hyperexcitability in pain transmission and an exaggerated release of excitatory amino acids and substance P [58]. Interestingly, an increase in cytokine concentrations (i.e., IL-1RA, IL-6, and IL-8) has been observed in patients with fibromyalgia [59]. Thus, both the increase in cytokines and the reduction in endogenous opioid receptors could be considered plausible explanations to support the role played by *COMT* (Met/Met genotype) in the severity of fatigue symptoms in fibromyalgia. However, further investigation is necessary to precisely determine the pathophysiological mechanisms underlying this relationship.

Some of our results contrast with previous findings. We observed that patients with fibromyalgia carrying the Met/Val genotype reported less fatigue than homozygous valine patients. Previous studies have indicated that patients with methionine alleles (i.e., Met/Met and Met/Val) are characterized by a greater severity of symptoms [11, 17, 34]. To solve these inconsistencies, some data have suggested that the effects of the *COMT* polymorphism on chronic clinical pain are highly dependent on the consequences derived from the pathology itself [60], such as psychiatric comorbidity or physical functioning. Regardless, it is important to keep in mind that neural communication does not just depend on the amount of neurotransmitter that is present in the synaptic cleft. It also depends on genes related to the reuptake or transport of these neurotransmitters that also influence synaptic transmission. Thus, analyses of other genes related to the reuptake and transport of catecholamines or haplotypes (which may have a greater effect on gene function than nonsynonymous variations [61]) could shed light on the inconsistencies found in our research with respect to other investigations.

As mentioned in the Results section, our data indicate that patients with fibromyalgia had worse scores than HC participants on most of the psychological questionnaires and self-reported pain and fatigue measures, except for the fear of pain outcomes. These results are in concordance with previous investigations showing high levels of depression, anxiety, or pain, among other symptoms in patients with fibromyalgia [13, 17, 18, 23, 34].

Given this, the question that arises is whether psychological symptoms, fatigue, or pain outcomes and *COMT* have any relationship to each other. Specifically, we tried to analyse whether psychological symptomatology (i.e., depression, anxiety, pain catastrophizing, fear of pain, or fear of movement) could have a moderating effect on self-reported fatigue or pain in genetically predisposed patients. We found that those patients whose genotype was associated with

lower enzymatic *COMT* activity (i.e., combining in a single group the Met carriers: the Met/ Met and Met/Val genotypes) and with higher levels of medical fear of pain, exhibited the highest self-reported fatigue. In contrast, this effect was not found in patients carrying the Val/Val genotype (associated with higher enzyme activity). In this line, several investigations have found that affective and cognitive factors can influence fatigue perception in both fibromyalgia [62] and chronic fatigue syndrome [63]. Specifically, high levels of fear of pain are usually related to more disability or pain [64]. Likewise, it seems that bad experiences in a medical context are good predictors of an increase in a medical fear of pain [65]. Given that fibromyalgia syndrome does not have a specific and effective treatment, patients may have increased feelings of discomfort, establishing ineffective ways to face pain (e.g., generating avoidance behaviours) [66], which may lead to the enhancement of fatigue perception. Therefore, it is not unreasonable to think that patients showing low enzymatic activity of *COMT* (i.e., Met allele carriers) could experience more feelings of fatigue, and this symptom could be exacerbated by the influence of emotional variables, such as their level of the fear of pain.

The relationship between *COMT* and self-reported fatigue, moderated by a fear of pain, deserves further consideration. We observed this novel relationship following the strategy of grouping the methionine carriers of the *COMT* gene (i.e., Met/Met + Met/Val genotypes) into a single group. This strategy has been carried out in previous experimental studies [15, 19, 23– 25]. It has been consistently reported that *COMT* methionine allele carriers have the lowest metabolic activity of the *COMT* enzyme [26]. In this sense, in patients with fibromyalgia, the methionine allele has been associated with an increased risk of suffering from fibromyalgia [19, 60]. More recently, a meta-analysis conducted by Lee and colleagues (2015) [15] revealed that patients with fibromyalgia carrying the Met allele of *COMT* (Met/Met + Val/Met genotypes) showed a higher impact of the disease than those carrying the Val allele. Our data seem to agree with such data suggesting that patients with fibromyalgia carrying the Met allele of the *COMT* gene would show a great severity of the syndrome, being especially sensitive to the influence of affective variables (i.e., fear of pain) on self-reported fatigue.

As previously mentioned, no differences in fear of pain scores were found between patients with fibromyalgia and healthy participants. In this sense, it may seem surprising to find that fear of pain acts as a moderator variable in the relationship between *COMT* genotypes and self-reported fatigue. However, it is important to keep in mind that the moderation analysis was only carried out within the patient group. Additionally, the lack of differences in fear of pain between groups may have different explanations. Previous research has reported that high levels of fear of pain have also been found in the general population [67, 68]. From a biological point of view, the presence of fear of pain in healthy people may be adaptive, since it could help the individual survive by avoiding possibly harmful or dangerous events. It would be interesting in future research to explore the relationship between these variables, linking them with neurobiological aspects.

Finally, with respect to the moderating effect of fear of pain on the relationship between the *COMT* gene and self-reported pain, some studies have indicated that patients with chronic pain carrying the *COMT* variants associated with lower enzyme activity and reporting higher levels of fear of pain had more intensified pain responses [69]. In our study, fibromyalgia Met carriers who had a higher medical fear of pain reported higher levels of pain. However, these analyses did not show significant differences, only a statistical trend. Several factors can be identified to explain the differences found between our results and previous results, such as the study carried out by George and colleagues (2015). First, they explored the relationship between pain responses and fear of pain in a sample of people suffering from shoulder pain (men and women). Second, they used a short version of the fear of pain questionnaire. As a final remark, the most relevant difference is that they analysed another SNP of the *COMT*

gene, rs6269, which also modulates the enzymatic activity of *COMT*. The combination of these differences could explain the presence of divergent results and make it difficult to perform a direct comparison with our results.

We acknowledge some limitations of our study. We only analysed a single SNP (Val158Met polymorphism). The analysis of haplotypes that are directly related to the enzymatic activity of *COMT* could be genetically more informative than the analysis of a single SNP. The concurrence of multiple haplotypes causing low *COMT* activity is more frequently associated with pain perception than the effect of a single SNP. Additionally, the relationship between genetics and the symptoms of chronic pain diseases is complex. Genetic factors do not act in isolation but interact with multiple other genetic and environmental factors that can increase the risk of suffering a greater severity of symptoms [34]. In this sense, it would be interesting to explore epigenetic factors in fibromyalgia that could explain the development of physical, affective, and cognitive symptoms. In future studies, it is necessary to increase the number of participants and analyse other *COMT* SNPs while trying to replicate the data of the present study.

## Conclusions

Our results showed that patients with fibromyalgia carrying the Met/Met genotype reported significantly higher levels of fatigue than heterozygote carriers and higher, but not significantly different fatigue scores from those carrying the Val/Val genotype. Thus, the present research is the first to report that different levels of fear of pain may act as a moderating factor in the relationship between the Met allele of the *COMT* gene and self-reported fatigue in fibromyalgia. The study of genetic polymorphisms would be a helpful tool for better identification and classification of these patients, allowing for more adequate and individualised multidisciplinary treatment. Psychological interventions (focused on affective symptomatology) might be useful not only for reducing psychological symptoms themselves but also for relieving fatigue and pain-related symptomatology, in which psychological symptoms may contribute as moderating factors in genetically predisposed patients with fibromyalgia.

## Supporting information

**S1 Table. Means and standard deviation (in parenthesis) of clinical measures of the *COMT* groups.** P-values of main effects of the *COMT* groups are also included.
(DOCX)

## Acknowledgments

The authors would like to thank all participants for taking part in the experiment.

## Author Contributions

**Conceptualization:** Francisco Mercado, Francisco Gómez-Esquer.

**Data curation:** David Ferrera, Francisco Mercado, Irene Peláez, Roberto Fernandes-Magalhaes, Paloma Barjola, Gema Díaz-Gil, Francisco Gómez-Esquer.

**Formal analysis:** David Ferrera, Francisco Mercado, David Martínez-Iñigo, Paloma Barjola, Carmen Écija, Gema Díaz-Gil, Francisco Gómez-Esquer.

**Funding acquisition:** Francisco Mercado.

**Investigation:** David Ferrera, Francisco Mercado, Irene Peláez, David Martínez-Iñigo, Paloma Barjola, Gema Díaz-Gil, Francisco Gómez-Esquer.

**Methodology:** David Ferrera, Francisco Mercado, David Martínez-Iñigo, Roberto Fernandes-Magalhaes, Paloma Barjola, Carmen Écija, Francisco Gómez-Esquer.

**Project administration:** Francisco Mercado.

**Resources:** Francisco Mercado, Francisco Gómez-Esquer.

**Software:** David Ferrera, David Martínez-Iñigo.

**Supervision:** David Ferrera, Francisco Mercado, Irene Peláez, Francisco Gómez-Esquer.

**Validation:** David Ferrera, Francisco Mercado, David Martínez-Iñigo, Francisco Gómez-Esquer.

**Visualization:** David Ferrera.

**Writing – original draft:** David Ferrera, Francisco Mercado, Irene Peláez, David Martínez-Iñigo, Roberto Fernandes-Magalhaes, Paloma Barjola, Carmen Écija, Gema Díaz-Gil, Francisco Gómez-Esquer.

**Writing – review & editing:** David Ferrera, Francisco Mercado, Irene Peláez, David Martínez-Iñigo, Roberto Fernandes-Magalhaes, Paloma Barjola, Carmen Écija, Gema Díaz-Gil, Francisco Gómez-Esquer.

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
