## [Decision Letter · Decision Letter 0]

3 Jul 2020

PONE-D-20-07498

Fear of pain moderates the relationship between fatigue perception and methionine allele of catechol-O-methyltransferase gene in fibromyalgia patients

PLOS ONE

Dear Dr. Ferrera,

Thank you for submitting your manuscript to PLOS ONE. After careful consideration, we feel that it has merit but does not fully meet PLOS ONE’s publication criteria as it currently stands. Therefore, we invite you to submit a revised version of the manuscript that addresses the points raised during the review process.

We look forward to receiving your revised manuscript.

Kind regards,

Shaheen E. Lakhan, MD, PhD, MEd, MS, FAAN

Academic Editor

PLOS ONE

Journal Requirements:

2. Please note that according to our submission guidelines (http://journals.plos.org/plosone/s/submission-guidelines), outmoded terms and potentially stigmatizing labels should be changed to more current, acceptable terminology.

For example: “Caucasian” should be changed to “white” or “of [Western] European descent” (as appropriate).

Reviewers' comments:

Reviewer's Responses to Questions

**Comments to the Author**

1. Is the manuscript technically sound, and do the data support the conclusions?

Reviewer #1: Partly

Reviewer #2: No

2. Has the statistical analysis been performed appropriately and rigorously? 

Reviewer #1: Yes

Reviewer #2: No

3. Have the authors made all data underlying the findings in their manuscript fully available?

Reviewer #1: Yes

Reviewer #2: Yes

4. Is the manuscript presented in an intelligible fashion and written in standard English?

Reviewer #1: Yes

Reviewer #2: Yes

5. Review Comments to the Author

Reviewer #1: Comments

Introduction

- The gene symbols need to be italicized.

- The discussion on how fatigue is important in this population should be added

- The reason for examining the association between this genotypes and fatigue should be included.

- There are a good number of studies investigate the association between COMT and fatigue in Irritable bowel syndrome, cancer, Chronic fatigue syndrome that could be used to support the relationship between the COMT and fatigue. This discussion should be included in the introduction

- 4 line 85, rechecked the statement. The previous section discussed about the relationships between the COMT genotypes and catastrophism.

Material and methods

- Participants: The ACR published several version of the diagnostic criteria. Give a specific year for the criteria used for this study.

- Add data collection procedure section to include the process on when and how the data (questionnaire and saliva) were collected

- Include the ethical consideration for clinical trials

- Include the variables and measures in the method session

Result:

- Line 120 to 147 should be in the result section

- Table 1 should be included in the result section

- Are there any potential impact of medication of the level of biomarkers?

- Line 272 typo on the figure number

Discussion

- Should include the unique finding of the association between the COMT and fatigue in this study (fibromyalgia) compared to evidence from other literature

- Are there any limitation of using the VAS (1 item) to measure fatigue in this study?

Reviewer #2: Fear of pain moderates the relationship between fatigue perception and methionine allele of catechol-O-methyltransferase gene in fibromyalgia patients

The authors describe an interesting study, in which they are aiming to relate COMT genotypes with psychological symptoms (e.g. catastrophizing, fear of pain) and physical outcomes (pain, fatigue). Their key findings are that 1) there are no differences in COMT genotypes across patients with fibromyalgia and controls; 2) there are no differences in psychological symptoms across COMT genotypes, but they do find interactions showing that Met/Met and Val/Val genotypes show higher fatigue compared to Met/Val; and 3) in a moderation analysis in fibromyalgia only, they found that fear of (medical) pain is a significant moderator for the relation between COMT genotype (Met/Met and Met/Val vs Val/Val) and fatigue in that only those patients with Met/Met or Met/Val AND higher fear of (medical) pain show increased fatigue.

Although the research questions are of interest to the field, the sample is decent and the study has potential to make an important contribution, I struggle with their some decisions in their analyses as well as their conclusions, which are not supported by the data (at least not by how the results are presented). I will explain my comments in more detail below in a way that I hope is useful for the authors.

Major points

The conclusions of the paper state that “fibromyalgia patients carrying the Met/Met genotype showed higher fatigue scores than patients carrying other genotypes.” According to their results though, both those carrying Met/Met as well as those with Val/Val showed higher fatigue scores compared to Met/Val. Thus, part of their conclusion as well as interpretations related to the lower COMT activity of Met/Met are not supported by the data, and maybe even misleading.

In addition, while they find differential effects for patients with Met/Val genotypes in terms of fatigue, they decided to group these together with Met/Met to represent ‘low activity’ in contrast to Val/Val (‘high activity’) in the moderation analyses. This does not make sense and can create a bias in the results. I would suggest to either always use the three groups or always take Met/Met and Met/Val together.

Other points

*I would rephrase ‘fibromyalgia patients’ into patients or individuals with fibromyalgia (across the entire manuscript) and stay away from the term ‘subjects’ altogether (and rather say ‘participants’).

*The manuscript is generally written in an understandable and clear manner, yet it would benefit from review by a native English speaker.

*The tables do not support the written results in that they do not contain the most relevant information. For instance, Table 3 presents differences across medication groups (which is only a control analysis), but differences across COMT groups are only presented in Supplement (while these are quite important).

*I think the authors mean ‘catastrophizing’ instead of ‘catastrophism’.

*I would have liked to see a bit more introduction or discussion on mechanisms underlying the effect of COMT; as in: how are catecholamines involved in pain perception, and potentially in affective or cognitive processes?

*It would be great to have a clearer idea about the study design and specific research questions at the end of the introduction.

Methods

*Sample size calculation. Was this done a priori, or more post-hoc to justify? If the authors really want to include this information, I would recommend also stating the predicted effect size, and a justification why the group sizes were unequal.

*Methods – line 106. “All participants were Caucasians”. Was that an inclusion criterion?

*What is the Hardy-Weinberg equilibrium? And would you not expect differences in frequencies across patients and controls, if COMT genotype is considered a risk factor? I would like to see some discussion on this.

Results

*Table 2 presents the allele frequencies as well. This is not explained in Methods, nor in Results. What does allele frequency entail/reflect, and does it provide different information than the genotype?

*If age differs, it should be included as a covariate in all analyses involving group comparisons.

*The order of results does not coincide with the order of the statistical analysis plan. This makes it a bit harder to follow.

*Line 232: “No main effect of COMT genotype on any of clinical variables was observed.” This is very relevant. Where are this data and the stats presented?

And were these part of ANOVA/GLM also including main effects of Group and Interaction COMT*Group? If so, these results need to be presented together, including all relevant statistics.

*Line 248: ‘the rest of the statistical contrasts’ – which ones are these? In general, I would recommend presenting all statistics, and not only the significant ones.

*Line 253: What are the ‘experimental effects’?

*Line 265: I think this is a typo and it should be physical symptoms as dependent variables (as introduced earlier). Although one can wonder whether self-reported pain and fatigue are ‘physical symptoms’; perhaps the authors can consider a different term.

*The authors conducted quite some moderation analyses – was there any correction for multiple testing? And were variables added in a stepwise fashion in the regression model?

*What exactly is presented in Table 4? It would be clearer if it included the standardized coefficients as well as relevant stats (including F/t and p values). And what was the total R2?

*I wonder whether pain and fatigue were highly correlated? And whether effects of fatigue would be explained by levels of pain.

*Table S1: it is unclear what the p-values refer to (which analysis).

Discussion

*As stated above, I do not agree that the conclusions are supported by the findings.

*I would urge the authors to rephrase the first parts of discussion where they discuss ‘modulation’ of symptoms by genotype. Modulation suggests some sort of causal and direct effect. Same holds for the abstract as a matter of fact. And for instance, ‘Met/Met modulate physical symptoms, but not psychological symptoms’ – how did the authors reach this conclusion?

*It is interesting that the moderation analyses are only significant for fear of (medical) pain, while this is the only variable that did not differ between individuals with fibromyalgia. I would love to see some discussion on this. Are findings perhaps not specific to this patient group? Would they find similar results in the control group?

*I miss discussion of relevance and implications of these findings. There is a tiny bit at end of abstract, but this is only about psychological symptoms, and does not take the interaction with genetics into account.

*Are there limitations to the use of saliva for DNA instead of blood?

Abstract

*Line 40: “The present data suggest that psychological factors are important in worsening the physical symptoms of fibromyalgia in genetically predisposed individuals” – besides that the data does not support this, there is also no information on progression over time.

6. PLOS authors have the option to publish the peer review history of their article (what does this mean?). If published, this will include your full peer review and any attached files.

Reviewer #1: No

Reviewer #2: No

---

## [Author Response · Author response to Decision Letter 0]

12 Aug 2020

Author's response to reviewers

ID PONE-D-20-07498

"Fear of pain moderates the relationship between fatigue perception and methionine allele of catechol-O-methyltransferase gene in fibromyalgia patients"

Authors:

David Ferrera (david.ferrera@urjc.es)

Francisco Mercado

Irene Peláez

David Martínez-Íñigo

Roberto Fernandes-Magalhaes

Paloma Barjola

Carmen Écija

Gema Díaz-Gil

Francisco Gómez-Esquer

Version: 2 Date: August 12, 2020

Dr. Shaheen E. Lakhan, MD, PhD, MEd, MS, FAAN

Academic Editor

PLOS ONE

Dear Dr. Shaheen,

Thank you very much for your and the reviewer´s helpful comments. We edited our manuscript following such comments and a point-by-point description of our changes is provided below. Changes in the manuscript are highlighted in red. We also provide a clean version of the manuscript. 

We believe that our manuscript has been considerably improved as a result of these revisions and hope that this revised version of the manuscript is acceptable for publication in PLOS ONE. Would like to thank you again for your consideration of our research and inviting us to submit the revised manuscript.

Best regards,

David Ferrera

• Reviewer 1

Introduction

1) The gene symbols need to be italicized. 

Response: Thank you for the suggestion. Following this recommendation, we have italicized the gene symbols across the entire manuscript.

2) The discussion on how fatigue is important in this population should be added

Response: Thank you for the reviewer’s comment. We have added in the Introduction section some considerations about the relationship between fatigue and fibromyalgia. See Page 4, (lines 74-82).

3) The reason for examining the association between this genotypes and fatigue should be included.

Response: We have made the suggested changes (see page 5, line 107-109).

4) There are a good number of studies investigate the association between COMT and fatigue in Irritable bowel syndrome, cancer, Chronic fatigue syndrome that could be used to support the relationship between the COMT and fatigue. This discussion should be included in the introduction

Response: Thank you for your suggestion. We have introduced some changes in the Introduction section to discuss about the relationship between COMT and fatigue in other diseases (see page 4, lines 82-88).

5) 4 line 85, rechecked the statement. The previous section discussed about the relationships between the COMT genotypes and catastrophism.

Response: According to the reviewer’s suggestions, we have edited the mentioned sentence and now this read as you can see in page 5, lines 107-109. 

Material and methods

6) Participants: The ACR published several versions of the diagnostic criteria. Give a specific year for the criteria used for this study.

Response: Thank you for your comment. Following this recommendation, we added the specific year for the fibromyalgia diagnostic criteria (see page 6, line 127).

7) Add data collection procedure section to include the process on when and how the data (questionnaire and saliva) were collected. 

Response: Following the reviewer´s suggestion we have introduced a new procedure section with information about the when and how the data were collected see page 6, line 148-167)

8) Include the ethical consideration for clinical trials.

Response: Although our study cannot be considered as a clinical trial, but a case-control study, we have included information on ethical aspects (see page 6, lines 143-145).

9) Include the variables and measures in the method section??

Response: Thank you for your comment. Following the reviewer´s suggestions, we have restructured the method section. We have included the information of measures and variables (see page 7, lines 153-164) and we have moved to the results section the results of the group effects on the clinical variables. 

Result:

10) Line 120 to 147 should be in the result section

Response: We have moved these lines from the Method section to the Results section (pages 10-11, line 242-254)

11) Table 1 should be included in the result section 

Response: Thank you for the reviewer’s comment. We have reorganized the Results section as recommended. As you can see, Table 1 has been moved in the Results section ((now this table is number 2, see pages 11-12, line 260)

12) Are there any potential impact of medication of the level of biomarkers?

Response: Thank you for the reviewer’s comment. Most of the individuals with fibromyalgia took analgesics or anti-inflammatories, benzodiazepines and antidepressants at the time of the experiment. 

Some of these drugs exert their action at the central nervous system level and, although none of them exert a direct effect on the biomarkers analysed in this study, we cannot rule out that an indirect effect may occur. For this reason, we have carried out different actions to minimize these effects.

Benzodiazepines (BZD) are a group of psychiatric drugs widely prescribed since their introduction in the clinical practice in the early 60's. The pharmacological action of BZD at molecular level over the Central Nervous System (CNS) is very well established. The mechanism of action of BZD is through the modulation of gamma-aminobutyric acid (GABA), an inhibitory neurotransmitter that suppresses the CNS (Rudolph, 2008). In general, BDZ are safe and effective for short-term treatment; however, long-term use is controversial due to the development of tolerance and their liability for physical dependence. All addictive drugs increase dopamine (DA) levels in the mesolimbic dopamine (DA) system (Tan, Rudolph, & Lüscher, 2011). In order to minimize this possible effect, in the present study participants discontinued BZD treatment 24 hours before conducting psychological tests.

In the case of antidepressant drugs, the main group of drugs prescribed to the participants in the studies were the selective serotonin receptor inhibitors (SSRIs). SSRIs antidepressants are the first-choice treatment in depression, in the anxiety disorder, the obsessive–compulsive disorder, the post–traumatic stress disorder, bulimia nervosa and the premenstrual dysphoric disorder. Depressed patients show less activity than normal of the serotonin neurotransmitter and the reuptake blockade at the site of the serotonergic presynaptic receptors 5HT1A, 5HT2C and 5HT3C increases neurotransmission in this system (Schloss & Williams, 1998). It is theorised that serotonin may inhibit the brain’s dopaminergic system, thus causing a reduction in dopamine activity. However, this reduced dopamine activity only seems to have a possible effect when simultaneous administration with antipsychotic drugs occurs, although clinical studies are inconclusive (Allsbrook, Fries, Szafara, & Regal, 2016). 

It is important to note that the present study has analysed the potential effects of medication on the clinical symptoms of patients:

Table 5 summarizes clinical data from patients according to the different type of medication taken (i.e. analgesics, benzodiazepines and antidepressants). ANOVAs did not reveal any significant effect on clinical symptomatology related to medications (all p- values were above 0.1), indicating that the severity of pain, fatigue or psychological symptoms was similar between the individuals with fibromyalgia who were taking medications and those who were not.

13) Line 272 typo on the figure number. 

Response: We have revised the sentence and on this line Figure 2 must be embedded as it appears.

Discussion

14) Should include the unique finding of the association between the COMT and fatigue in this study (fibromyalgia) compared to evidence from other literature

Response: Thank you for this helpful suggestion. We have described the findings of the association of COMT and fatigue in fibromyalgia and we have compared these results to other published evidence in the Discussion section, as recommended. We have edited the text and included a new paragraph on page 17-18, lines 349-376.

15) Are there any limitation of using the VAS (1 item) to measure fatigue in this study?

Response: Thank you for your question. The use of visual analog scales (VAS) has been widely used in several studies in different pathologies (Blesch et al., 1991; Crawford, Piault, Lai, & Bennett, 2011; Leung, Chan, Lee, & Lam, 2004). Consistently, the results using a VAS show a high correlation with other tests measuring fatigue such as the Profile of Mood States (POMS) or Stanford Sleepiness Scale (Brunier & Graydon, 1996; K. A. Lee, Hicks, & Nino-Murcia, 1991). Likewise, VAS have been reported to have a good capability to discriminate cases from non-cases with an acceptable degree of specificity and sensitivity, as well as moderate psychometric properties (Whitehead, 2009). Also, in the case of fibromyalgia, the utility of this tool has been evaluated (Crawford et al., 2011). In this study, the authors report that the VAS has a high correlation with the fatigue-related items of the Fibromyalgia Impact Questionnaire (FIQ) and the SF-36 Vitality Scale, also supporting the use of VAS for assessment fatigue in fibromyalgia in experimental studies.

• Reviewer 2

The authors describe an interesting study, in which they are aiming to relate COMT genotypes with psychological symptoms (e.g. catastrophizing, fear of pain) and physical outcomes (pain, fatigue). Their key findings are that 1) there are no differences in COMT genotypes across patients with fibromyalgia and controls; 2) there are no differences in psychological symptoms across COMT genotypes, but they do find interactions showing that Met/Met and Val/Val genotypes show higher fatigue compared to Met/Val; and 3) in a moderation analysis in fibromyalgia only, they found that fear of (medical) pain is a significant moderator for the relation between COMT genotype (Met/Met and Met/Val vs Val/Val) and fatigue in that only those patients with Met/Met or Met/Val AND higher fear of (medical) pain show increased fatigue.

Although the research questions are of interest to the field, the sample is decent and the study has potential to make an important contribution, I struggle with their some decisions in their analyses as well as their conclusions, which are not supported by the data (at least not by how the results are presented). I will explain my comments in more detail below in a way that I hope is useful for the authors.

[REPLY]

Thank you very much the reviewer for the constructive comments and helpful suggestions. We fundamentally agree with the comments made by the reviewer, and we have included the corresponding changes into the manuscript. Additionally, we have carefully revised the manuscript to ensure that the text is optimally phrased and free from typographical and grammatical errors. Manuscript was professionally proofread. Thus, we provide a detailed point-by-point description of the changes we made in response to reviewer’s suggestions. 

Major points

16) The conclusions of the paper state that “fibromyalgia patients carrying the Met/Met genotype showed higher fatigue scores than patients carrying other genotypes.” According to their results though, both those carrying Met/Met as well as those with Val/Val showed higher fatigue scores compared to Met/Val. Thus, part of their conclusion as well as interpretations related to the lower COMT activity of Met/Met are not supported by the data, and maybe even misleading.

Response: Thank you for the helpful comments. In order to make the conclusions more adequately and accurately reflect the data reported in the study, we have generally modified the conclusions both in the abstract and in the final part of the manuscript. Likewise, in the discussion section (see page 21, lines 455-468), we discuss the unexpected results of the Val/Val genotype and try to explain these based on previous literature. 

17) In addition, while they find differential effects for patients with Met/Val genotypes in terms of fatigue, they decided to group these together with Met/Met to represent ‘low activity’ in contrast to Val/Val (‘high activity’) in the moderation analyses. This does not make sense and can create a bias in the results. I would suggest to either always use the three groups or always take Met/Met and Met/Val together.

Response: Thank you for your comment. There are many studies in which Val/Met (GA) and Met/Met (AA) genotypes are grouped into a single group. This procedure is justified because Met allele it is responsible for the lowest metabolic activity of the COMT enzyme. In fact, the Val-158-Val genotype generates a fully effective enzyme, but the Met-158-Val or Met-158-Met produces an intermediate-activity or a defective enzyme, respectively.

In 2015, Lee at al. (Y. H. Lee, Kim, & Song, 2015), conducted a meta-analysis of the associations of the COMT Val158Met polymorphism with fibromyalgia risk as well as FIQ (Fibromyalgia Impact Questionnaire) score in individuals with fibromyalgia. A total of 1531 fibromyalgia patients from 15 studies were included in this meta-analysis. The meta-analysis revealed an association between fibromyalgia and the COMT Met/Met + Val/Met genotype in all study subjects.

Moreover, it has been postulated that people with a COMTLL (Met/Met) or COMTLH (Met/Val) genotype were less able to tolerate pain than persons with a COMTHH (Val/Val) genotype. Positron emission tomography scans showed that this occurred because people with COMTLL (Met/Met) or COMTLH (Met/Val) made less beta-endorphin (Berthele et al., 2005).

Other studies in which Val/Met (GA) and Met/Met (AA) genotypes are grouped into a single group:

Hooten, W. M., Biernacka, J. M., OʼBrien, T. G., Cunningham, J. M., & Black, J. L. (2019). Associations of catechol-O-methyltransferase (rs4680) single nucleotide polymorphisms with opioid use and dose among adults with chronic pain. Pain, 160(1), 263–268. https://doi.org/10.1097/j.pain.0000000000001400

The COMT Val158Met Polymorphism and Reaction to a Transgression: Findings of Genetic Associations in Both Chinese and German Samples Front. Behav. Neurosci., 03 August 2018 | https://doi.org/10.3389/fnbeh.2018.00148.

Likewise, we have carried out several moderation analyses dividing the COMT genotypes into three levels (Met/Met, Met/Val and Val/Val). Analyzes reveal a clear trend toward significance in the regression coefficient for fatigue (β = -0.212; p = 0.056). Also, the fatigue regression equation was estimated for each group and showed that patients carrying Met/Val genotypes and scoring higher in medical fear of pain reported more elevated levels of fatigue (β = 0.333; p = 0.024). In the case of carriers of the Met/Met genotype, the results show a clear trend towards significance (β = 0.425; p = 0.055), following the same direction as the results for the Met/Val genotype, that is, those subjects who report more medical fear of pain they will report more fatigue. In those patients carrying the Val/Val genotype, medical fear of pain outcomes did not predict differences in fatigue (β = - 0.134; p = 0.531) for this group. 

Other points

18) I would rephrase ‘fibromyalgia patients’ into patients or individuals with fibromyalgia (across the entire manuscript) and stay away from the term ‘subjects’ altogether (and rather say ‘participants’). 

Response: Thank you for the reviewer’s suggestion. We have rephrased the term fibromyalgia patients into individuals with fibromyalgia or patients across the entire manuscript.

19) The manuscript is generally written in an understandable and clear manner, yet it would benefit from review by a native English speaker.

Response: Thank you for your suggestion. in order to improve and make the paper more understandable, the manuscript was professionally proofread.

20) The tables do not support the written results in that they do not contain the most relevant information. For instance, Table 3 presents differences across medication groups (which is only a control analysis), but differences across COMT groups are only presented in Supplement (while these are quite important).

Response: Following the reviewer’s suggestion, we have changed various tables in the entire manuscript. We have introduced a new table with the information on the effects of COMT x Group Interaction (previously it was a table of supplementary material). In this revised version of the manuscript we renamed this as Table 3 see page 12, line 282). Regarding the table 3 that summarizes the statistics of the medication control analysis, we have changed it by placing it at the end of the results section (now this table is number 5, see page 15-16, line 330)

21) I think the authors mean ‘catastrophizing’ instead of ‘catastrophism’.

Response: Following the reviewer's recommendations, we have revised the manuscript to unify the use of these terms. We have used the term pain catastrophizing in the entire manuscript to refer to that variable. 

22) I would have liked to see a bit more introduction or discussion on mechanisms underlying the effect of COMT; as in: how are catecholamines involved in pain perception, and potentially in affective or cognitive processes?

Response: Thank you for this helpful suggestion. We have added information about the relationship between COMT genotypes and pain perception in the Introduction section (see pages 3-4, lines 69-73). Furthermore, the relationship between COMT and affective processing (see page 5, lines 100-106).

23) It would be great to have a clearer idea about the study design and specific research questions at the end of the introduction.

Response: Thank you for your suggestion. We have changed the end of the Introduction section in order to clarify the study design and the research questions. Now, this part of the Introduction section reads as follows: ´Therefore, the aim of the present case-control study was to examine differences on pain, fatigue and other psychological outcomes (depression, anxiety, fear of pain, pain catastrophizing and fear of movement) in fibromyalgia due to the COMT gene. On the other hand, we wonder if patients with fibromyalgia carrying a certain genotype will have higher levels of self-reported fatigue and pain, if they also have higher levels of self-reported psychological symptoms. For this purpose, we analyzed the Val158Met polymorphism of the COMT gene in individuals with fibromyalgia and healthy participants and their scores in pain, fatigue and psychological tests´.

Methods

24) Sample size calculation. Was this done a priori, or more post-hoc to justify? If the authors really want to include this information, I would recommend also stating the predicted effect size, and a justification why the group sizes were unequal.

Response: Thank you for your question. The post-hoc calculation of statistical power has been used to justify the necessary sample size. To clarify this point we have modified the last part of the method (page 9-10, lines 222-225). In addition to reporting the power of statistically significant ANOVAs, we have provided the statistical power of moderation analysis that showed significant differences. As can be seen in the current version of the manuscript, all statistical analysis yielded a power greater than 0.8 and, therefore, a high statistical power.

25) Methods – line 106. “All participants were Caucasians”. Was that an inclusion criterion?

Response: Thank you for your question. Indeed, ethnicity is an inclusion criterion since it has been reported that the differences in results between investigations that explore fibromyalgia and genetic factors may be due to the ethnic origin of the different samples used, where a strong association has been found (C. Lee, Liptan, Kantorovich, Sharma, & Brenton, 2018). In this sense, these authors have reported that non-Caucasian women (African Americans and Hispanics) are at increased risk for FM. In this way, we have modified the sentence on page 6, lines 137, in the Method section, to clearly specify the inclusion criteria used.

26) What is the Hardy-Weinberg equilibrium? And would you not expect differences in frequencies across patients and controls, if COMT genotype is considered a risk factor? I would like to see some discussion on this.

Response: Thank you for your comments. The Hardy-Weinberg equilibrium determines which frequencies must be observed in the population for each genotype based on the frequencies of the alleles. Under normal conditions, if the transmission of alleles from parents to offspring is independent, the probability of observing a specific combination of alleles depends on the product of the probabilities of each allele. Before carrying out an association study, it must be verified if the balance is met in the control sample. It can also be done in the patients’ sample and it is possible that the equilibrium is not met, as the reviewer has commented. It may be indicative that this gene can be associated with the development of the disease (Iniesta, Guinó, & Moreno, 2005). 

In the case of fibromyalgia syndrome, multiple studies have been carried out that have attempted to relate the frequency of genotypes with the risk of suffering from fibromyalgia, finding different results. While some of them have reported an association of the Met/Met genotype with the risk of the disease (Gürsoy et al., 2003; Inanir et al., 2014; Y. H. Lee et al., 2015; Park et al., 2016), others failed to find significant results (Barbosa et al., 2012; Estévez-López et al., 2018; Hatami et al., 2020; Zhang, Zhu, Chen, & Zhao, 2014).

In this sense, fibromyalgia has been defined as a multifactorial syndrome (Wolfe et al., 2010), whose pathophysiology involves a large number of factors, including abnormalities in the neuroendocrine system and in the autonomic nervous system, genetic factors, psychological variables, and environmental stressors (Bradley, 2009). Specifically, twin studies suggest that approximately 50% of the risk of fibromyalgia and other related symptoms is genetic, this informs us that the other 50% can be explained by environmental factors (Kato, Sullivan, Evengård, & Pedersen, 2009). Among these environmental factors, we find early exposure to stressors, such as maternal cortisol levels, maternal pain or deprivation, and physical or substance abuse can influence the development of neurobiological or psychological disorders such as those seen in the fibromyalgia (Low & Schweinhardt, 2012). On the other hand, physical stressors such as stressors in the workplace or prolonged load with excessive weight, have also been associated with the development of the patient (Bradley, 2009). Finally, the influence of age, sex or ethnicity has been reported as relevant to explain the variability in the results found in the risk of suffering from fibromyalgia) (C. Lee et al., 2018).

All these data lead us to conclude that fibromyalgia is a syndrome that is probably not due to a single factor but to the interaction of a variable number of these (genetic factors that increase sensitivity to pain, stress in adulthood or early age, influence of sex hormones, etc.,) that generate a phenotype with a higher risk of developing chronic pain (Bradley, 2009). 

Results

27) Table 2 presents the allele frequencies as well. This is not explained in Methods, nor in Results. What does allele frequency entail/reflect, and does it provide different information than the genotype?

Response: Thank you for your comment. The use of the frequency of alleles does not provide different or more relevant information than the frequency of genotypes can provide. Despite this, the inclusion of this information has been done because it helps a better follow-up in the description of the characteristics of the sample.

28) If age differs, it should be included as a covariate in all analyses involving group comparisons.

Response: Thank you for your reviewer´s comment. As you can see in page 13, lines 284-290 we have reported various ANCOVAs in which explore the possible influence of age in the results. The ANCOVAs carried out to control for the possible effect of age on the clinical symptoms revealed that the influence of the COMT genotypes on clinical outcomes are independent of age reported by each group (individuals with fibromyalgia and HC).

29) The order of results does not coincide with the order of the statistical analysis plan. This makes it a bit harder to follow.

Response: Thank you for the reviewers’ comment. In order to make these parts more understandable we have included several changes to make easier to follow these sections. 

30) Line 232: “No main effect of COMT genotype on any of clinical variables was observed.” This is very relevant. Where are this data and the stats presented?

Response: Thank you. Indeed, in the manuscript the statistics referring to the main effects of genotype do not appear. Following the reviewer´s suggestion and in order to clarify this information, we have introduced a table as supplementary material (S1 table), showing the means, standard deviations and p-value of the main effects of genotype.

31) And were these part of ANOVA/GLM also including main effects of Group and Interaction COMT*Group? If so, these results need to be presented together, including all relevant statistics.

Response: Thank you for the reviewers’ suggestion. We have edited this part of the Results section including main effects of Group, COMT and Interaction effects COMT x Group, as you can see in pages 10-13, lines 242-90 In the case of the Interaction effects COMT x Group, we have changed the supplementary material (where the means, standard deviations and p-values of the interaction effects were presented) for a table within the manuscript (now this information can be seen in Table 3). 

32) Line 248: ‘the rest of the statistical contrasts’ – which ones are these? In general, I would recommend presenting all statistics, and not only the significant ones.

Response: Thank you for your suggestion. This information now it presents in the Table 3

33) Line 253: What are the ‘experimental effects’?

Response: Thank you for your comment. We have revised the mentioned statements in order to make them clearer and clarify its meaning. Now it reads as follows: ´Furthermore, ANCOVAs carried out to control for the possible effect of age on the clinical symptoms revealed that the influence of the COMT genotypes on clinical outcomes are independent of age reported by each group (individuals with fibromyalgia and HC)’.

34) Line 265: I think this is a typo and it should be physical symptoms as dependent variables (as introduced earlier). Although one can wonder whether self-reported pain and fatigue are ‘physical symptoms’; perhaps the authors can consider a different term. 

Response: Thank you for your comment. We have removed the typo. The sentence has been rewritten as follows: ´Here, COMT genotypes were introduced as an independent variable, psychological scores test as a moderation variable and self-reported pain or fatigue as dependent variables´.

On the other hand, we have considered using another term instead of physical symptoms, because both fatigue and pain have an important psychological component. We have decided to change that term and refer to the two variables as self-reported pain or self-reported fatigue or just pain and fatigue.

35) The authors conducted quite some moderation analyses – was there any correction for multiple testing? And were variables added in a stepwise fashion in the regression model?

Response: Thank you for your comment. Authors understand the reviewers’ concerns on the effects of multiple comparisons in the risk of increasing the type I error. To reduce this possibility, regression analysis was conducted using the stepwise procedure. In addition to the p-value, increases in R square are now reported to show the contribution of the interaction term to the explanation of the DV.

36) What exactly is presented in Table 4? It would be clearer if it included the standardized coefficients as well as relevant stats (including F/t and p values). And what was the total R2?

Response: This table presents the information related to the moderation analysis. In order to make the information in this table clearer, we have revised the Table 4 and we have included the necessary statistics. We have also included the total R2 (see page 15, line 318).

37) *I wonder whether pain and fatigue were highly correlated? And whether effects of fatigue would be explained by levels of pain.

Response: Thank you for your questions. We have reanalysed these variables. Pain and fatigue are highly correlated in our sample (r = 0.781, p = 0.001), in such a way that we have introduced pain as a covariable in an ANCOVA, in which the dependent variable was fatigue and it have two factors (genotypes of the COMT and group). The effects remain significant for the interaction COMT and Group (F (2,151) = 3.590, p = 0.030), for the main effect of group (F (2,151) = 8.289, p = 0.005), but still do not show significant differences for the main effects of genotype (F (2,151) = 2.060, p = 0.578). Therefore, we can conclude that the differences in fatigue between genotypes and groups are independent of the effects of pain.

38) *Table S1: it is unclear what the p-values refer to (which analysis).

Response: Following this comment and previous ones from this reviewer, we have introduced this supplementary material as a table within the manuscript (see, Table 3). This table shows the means and standard deviations of the clinical variables. The information is shown separately by group (fibromyalgia and healthy control) and genotype (Val / Val, Met / Val and Met / Met). In addition, the p-values for the COMT x Group interaction effects are reported.

Discussion

39) As stated above, I do not agree that the conclusions are supported by the findings.

Response: As we have previously commented, following the useful recommendations of the reviewer, we have modified and reformulated the conclusions so that they fit more appropriately to our results. These changes have been made throughout both in the Conclusion section and in the Abstract (page 2, lines 41-44), and in the conclusion section of the manuscript (see page 21, lines 455-468). 

40) I would urge the authors to rephrase the first parts of discussion where they discuss ‘modulation’ of symptoms by genotype. Modulation suggests some sort of causal and direct effect. Same holds for the abstract as a matter of fact. And for instance, ‘Met/Met modulate physical symptoms, but not psychological symptoms’ – how did the authors reach this conclusion?

Response: Thank you for this helpful suggestion. We have reformulated the discussion and the abstract section to replace the term modulation by other terms more adjusted to our data. On the other hand, we have also modified the phrase suggested by the reviewer and it reads as follows: ‘our results are the first, according to our knowledge, to show that patients carrying the Met/Met genotype (i.e. identified as the low COMT activity genotype) exhibited significantly higher levels of self-reported fatigue than heterozygotes carriers and worse-but not significant- scores than Val/Val subgroup‘.

41) It is interesting that the moderation analyses are only significant for fear of (medical) pain, while this is the only variable that did not differ between individuals with fibromyalgia. I would love to see some discussion on this. Are findings perhaps not specific to this patient group? Would they find similar results in the control group?

Response: The two analysis that the reviewer points out in this comment have been carried out with different methods. While ANOVAs have been performed to observe differences between the two groups (fibromyalgia and healthy participants), that is, two different groups are compared; the moderation analysis was carried out only within the fibromyalgia group, that is, using a single group. Therefore, the two analysis inform us of different aspects of the possible influence genotypes of the COMT gene on characteristic symptoms in fibromyalgia. On one hand, analyzing whether the scores of the patients are comparable to that of the healthy participants or are related to some specific genotype. On the other hand, the analysis related to whether within the fibromyalgia group, one allele or another has a significant relationship with psychological symptoms and self-reported pain or fatigue. 

Despite this, we have considered the reviewer's proposal as to whether the results may not be specific to the group of patients with fibromyalgia very interesting. For this we have carried out several moderation analyses, only with the sample of healthy participants. The analysis conducted to investigate potential moderation effect of medical fear of pain in the relationship between the COMT gene and self-reported pain or fatigue in healthy controls participants, did not show a significant interaction effect (for self- eported pain: β = 0.241; p = 0.475; for self-reported fatigue: β = 0.129; p = 0.708). Therefore, the data seems to show that these results are specific to the group of participants with fibromyalgia.

42) I miss discussion of relevance and implications of these findings. There is a tiny bit at end of abstract, but this is only about psychological symptoms, and does not take the interaction with genetics into account.

Response: Thank you for the suggestion. Following the reviewer’s comment, we have included edited the manuscript including a reference to the implications of the present results. This information has been provided at the end of the manuscript (see page 21, lines 461-464)

43) Are there limitations to the use of saliva for DNA instead of blood? 

Response: Thank for the reviewer´s question. Various studies have shown that the quality and quantity DNA from saliva and blood samples is comparable when genotyping using either Taqman assays or genome-wide chip arrays. Furthermore, saliva, as compared to blood collection, has the following advantages: it requires no specialized personnel for collection, allows for remote collection by the patient, is painless, well accepted by participants, has decreased risks of disease transmission, does not clot, can be frozen before DNA extraction and possibly has a longer storage time.

Abraham, J.E., Maranian, M.J., Spiteri, I. et al. Saliva samples are a viable alternative to blood samples as a source of DNA for high throughput genotyping. BMC Med Genomics 5, 19 (2012). https://doi.org/10.1186/1755-8794-5-19

Bahlo, M., Stankovich, J., Danoy, P., Hickey, P. F., Taylor, B. V., Browning, S. R., Australian and New Zealand Multiple Sclerosis Genetics Consortium (ANZgene), Brown, M. A., & Rubio, J. P. (2010). Saliva-derived DNA performs well in large-scale, high-density single-nucleotide polymorphism microarray studies. Cancer epidemiology, biomarkers & prevention: a publication of the American Association for Cancer Research, cosponsored by the American Society of Preventive Oncology, 19(3), 794–798. https://doi.org/10.1158/1055-9965.EPI-09-0812

Abstract

44) *Line 40: “The present data suggest that psychological factors are important in worsening the physical symptoms of fibromyalgia in genetically predisposed individuals” – besides that the data does not support this, there is also no information on progression over time.

Response: Thanks for the reviewer´s suggestion. We have introduced some changes in this paragraph belonging to the Abstract to better specify the conclusion of the research (see, page 2, lines 41-44)

---

## [Decision Letter · Decision Letter 1]

18 Dec 2020

PONE-D-20-07498R1

Fear of pain moderates the relationship between fatigue perception and methionine allele of catechol-O-methyltransferase gene in fibromyalgia patients

PLOS ONE

Dear Dr. Ferrera,

Thank you for submitting your manuscript to PLOS ONE. After careful consideration, we feel that it has merit but does not fully meet PLOS ONE’s publication criteria as it currently stands. Therefore, we invite you to submit a revised version of the manuscript that addresses the points raised during the review process.

Thank you for addressing the majority of the comments raised by the Reviewers, however, please  address  comments and suggestions  of Reviewer 2

We look forward to receiving your revised manuscript.

Kind regards,

Mahmoud Abu-Shakra, MD

Academic Editor

PLOS ONE

Reviewers' comments:

Reviewer's Responses to Questions

**Comments to the Author**

1. If the authors have adequately addressed your comments raised in a previous round of review and you feel that this manuscript is now acceptable for publication, you may indicate that here to bypass the “Comments to the Author” section, enter your conflict of interest statement in the “Confidential to Editor” section, and submit your "Accept" recommendation.

Reviewer #2: (No Response)

Reviewer #3: (No Response)

2. Is the manuscript technically sound, and do the data support the conclusions?

Reviewer #2: Partly

Reviewer #3: Yes

3. Has the statistical analysis been performed appropriately and rigorously? 

Reviewer #2: No

Reviewer #3: Yes

4. Have the authors made all data underlying the findings in their manuscript fully available?

Reviewer #2: No

Reviewer #3: Yes

5. Is the manuscript presented in an intelligible fashion and written in standard English?

Reviewer #2: Yes

Reviewer #3: Yes

6. Review Comments to the Author

Reviewer #2: “Fear of pain moderates the relationship between fatigue perception and methionine allele of catechol-O-methyltransferase gene in fibromyalgia patients”

The authors have worked hard on addressing the suggestions, questions and revising the manuscript accordingly. As a result, the manuscript greatly improved. Although I am quite satisfied with most points, but I do have some remaining concerns and questions, which I will address below. In addition, some discussion that is in the response letter would make valuable contributions to the paper.

Point 16 - The main conclusion is now phrased a bit more careful, but still confusing: “Main results indicated that patients carrying the Met/Met genotype reported significantly higher levels of fatigue compared to heterozygotes carriers and higher, but not significant, scores than Val homozygotes carriers.” I think this should be: “and higher, but not significantly different from Val homozygote carriers” – or something along this line. Note that addressing my next point could actually also be relevant for this point (if it turns out that Val/Val carriers do show higher fatigue and pain compared to the other groups combined).

Point 17 – I appreciate the explanation and do understand the rationale for taking Met/Met and Val/Met together (representing the Val158Met polymorphism carriers if I understand it correctly now). However, the use of two different approaches in this manuscript is still problematic if this is not explicitly explained nor addressed.

If the authors want to keep both strategies (use 3 genotype groups for part A-group differences, and use 2 genotype groups for part B-moderation), it should be a) explicitly acknowledged that two different groupings are used (3 groups versus 2 groups) and why based on provided references e.g., and b) the 2 genotype groups should be formally (statistically) compared to see if results hold up when using this different grouping (i.e., to see if Val/Val carriers indeed show higher self-reported pain and fatigue than the other group) in order to justify using this alternative grouping for part B-moderation.

The current strategy is problematic as the moderation analyses build on the group difference findings, but in fact, they cannot be built on this due to different analytical strategies. And the analyses will be invalid.

The alternative would be that the analyses that are described in the response letter are presented in the paper as a way to justify using the different groupings.

Point 18 - The authors have changed wording to ‘patients/individuals with fibromyalgia’ and avoided ‘subjects’ for the most part. Yet, I would recommend them to do another check as some old terms are remaining (e.g., line 103, 354 or 421) and also the title still uses the old style. Also, talking about the title: I am not sure ‘fatigue perception’ is an appropriate term in this case.

The same with ‘catastrophism’ which is still used in the manuscript.

Tables - The authors did a good job updating the result sections and corresponding tables.

Point 26 – It is still unclear how the Hardy-Weinberg equilibrium was tested. What was the expected distribution? And also, some of this discussion in the response letter would merit inclusion in the manuscript.

Point 37 – Wonderful. This would make an excellent contribution to the manuscript, so I would suggest including it.

Point 41 – This is a nice discussion that also deserves to be added to the manuscript. Because no matter how you look at it, the one (and only) variable that is not elevated in the patients compared to controls does act as a moderator variable (even though analysis strategies are different). Discussion on why this would be, or at least acknowledging this, is very relevant.

General point - One of my struggles was with the use of terms such as genotype, (Methionine) alleles, polymorphisms, heterozygotes carriers, Val158Met, etc of which some seem to be synonyms, but others mean slightly different things.

Many things have been clarified in the responses as well as in the manuscript, for which I am grateful, but it would greatly improve the comprehensiveness of the manuscript if there was some consistency in referring to some of these jargon terms, especially in a journal like this which is read by a broad audience.

Minor points

*Abstract line 33 “influences of COMT genotypes and group of participants” – does the latter refer to comparing patients with controls? Please rephrase.

*Page 6, lines 138-142 – This is not the place for this information, this should rather be placed at the description of the sample (around Table 2 e.g.)

*Page 12, lines 265-266 – Please include the comparison Met/Met and Val/Val

*Page 12, line 268 – Please avoid qualifying words such as a ‘strong’ trend.

*Page 12 – It is unclear whether there were main effects of COMT gene (across groups).

Reviewer #3: I enjoyed reading the paper, the authors went through an interesting path tying psychological aspects with genetics, the paper is well written abd balanced.

7. PLOS authors have the option to publish the peer review history of their article (what does this mean?). If published, this will include your full peer review and any attached files.

Reviewer #2: No

Reviewer #3: **Yes: **Howard AMITAL

---

## [Author Response · Author response to Decision Letter 1]

8 Feb 2021

Author's response to reviewers

ID PONE-D-20-07498R1

"Fear of pain moderates the relationship between self-reported fatigue and methionine allele of catechol-O-methyltransferase gene in patients with fibromyalgia"

Authors:

David Ferrera (david.ferrera@urjc.es)

Francisco Mercado

Irene Peláez

David Martínez-Íñigo

Roberto Fernandes-Magalhaes

Paloma Barjola

Carmen Écija

Gema Díaz-Gil

Francisco Gómez-Esquer

Version: 3 Date: February 1, 2021

Dr. Mahmoud Abu-Shakra, MD

Academic Editor

PLOS ONE

Dear Dr. Abu-Shakra,

Thank you very much for your e-mail and comments submitted by the reviewer #2. We have responded to the reviewer's comments and suggestions point by point and we hope that this new version is acceptable for publication in PLOS ONE. We would like to thank you again for your consideration of our research and inviting us to submit the revised manuscript.

Best regards,

David Ferrera

• Reviewer #2

Point 16 - The main conclusion is now phrased a bit more careful, but still confusing: “Main results indicated that patients carrying the Met/Met genotype reported significantly higher levels of fatigue compared to heterozygotes carriers and higher, but not significant, scores than Val homozygotes carriers.” I think this should be: “and higher, but not significantly different from Val homozygote carriers” – or something along this line. Note that addressing my next point could actually also be relevant for this point (if it turns out that Val/Val carriers do show higher fatigue and pain compared to the other groups combined).

Response: Thank you for the suggestion. Following this recommendation, we rephrased the Abstract (see page 2, lines 39) and the Conclusion (page 23-24, line 513-514) sections in order to make them more understandable.

Point 17 – I appreciate the explanation and do understand the rationale for taking Met/Met and Val/Met together (representing the Val158Met polymorphism carriers if I understand it correctly now). However, the use of two different approaches in this manuscript is still problematic if this is not explicitly explained nor addressed.

If the authors want to keep both strategies (use 3 genotype groups for part A-group differences, and use 2 genotype groups for part B-moderation), it should be a) explicitly acknowledged that two different groupings are used (3 groups versus 2 groups) and why based on provided references e.g., and b) the 2 genotype groups should be formally (statistically) compared to see if results hold up when using this different grouping (i.e., to see if Val/Val carriers indeed show higher self-reported pain and fatigue than the other group) in order to justify using this alternative grouping for part B-moderation.

The current strategy is problematic as the moderation analyses build on the group difference findings, but in fact, they cannot be built on this due to different analytical strategies. And the analyses will be invalid.

The alternative would be that the analyses that are described in the response letter are presented in the paper as a way to justify using the different groupings.

Response: Thank you for your comments. We agree with the reviewer that the use of both strategies might be confusing. However, this approach combining the use of two analysis strategies has been regularly used to explore the relationship between genetic factors and clinical/cognitive variables in pain. Some articles where this type of analysis has been carried out are the following: 

-Cargnin, S., Magnani, F., Viana, M., Tassorelli, C., Mittino, D., Cantello, R., Sances, G., Nappi, G., Canonico, P. L., Genazzani, A. A., Raffaeli, W., & Terrazzino, S. (2013). An Opposite-Direction Modulation of the COMT Val158Met Polymorphism on the Clinical Response to Intrathecal Morphine and Triptans. The Journal of Pain, 14(10), 1097–1106. https://doi.org/10.1016/j.jpain.2013.04.006

-Cohen, H., Neumann, L., Glazer, Y., Ebstein, R. P., & Buskila, D. (2009). The relationship between a common catechol-O-methyltransferase (COMT) polymorphism val158met and fibromyalgia. Clinical and Experimental Rheumatology, 27(5), S51–S56.

-Gürsoy, S., Erdal, E., Herken, H., Madenci, E., Alaşehirli, B., & Erdal, N. (2003). Significance of catechol-O-methyltransferase gene polymorphism in fibromyalgia syndrome. Rheumatology International, 23(3), 104–107. https://doi.org/10.1007/s00296-002-0260-5

-Lee, Y. H., Kim, J.-H., & Song, G. G. (2015). Association between the COMT Val158Met polymorphism and fibromyalgia susceptibility and fibromyalgia impact questionnaire score: a meta-analysis. Rheumatology International, 35(1), 159–166. https://doi.org/10.1007/s00296-014-3075-2

-Sindermann, C., Luo, R., Zhang, Y., Kendrick, K. M., Becker, B., & Montag, C. (2018). The COMT Val158Met Polymorphism and Reaction to a Transgression: Findings of Genetic Associations in Both Chinese and German Samples. Frontiers in Behavioral Neuroscience, 12. https://doi.org/10.3389/fnbeh.2018.00148

Although we have decided to keep the original strategy, several changes have been introduced in different parts of the manuscript, as recommended by the reviewer's suggestions. Specifically, we have added a theoretical explanation to justify why this approach might be a valid grouping strategy, being supported by a well-established physiological explanation (see pages 3-4, lines 72-80). On the other hand, we conducted a new moderation analysis (Results section). to explore the role of the psychological symptoms as potential factors to moderate the relationship between pain/fatigue and the three COMT genotypes (i.e., Met/Met, Met/Val and Val/Val) within the sample of patients with fibromyalgia (see pages 15, lines 324-329). The results derived from these analyses did not show any significant differences. Thus, we kept the grouping strategy including methionine carriers into a single group (Met/Met, Met/Val) and Valine homozygotes (Va/Val) in a different one. Moreover, the use of this strategy has recently been recommended in fibromyalgia. Lee and colleagues (2015) explored the associations between the Val158Met polymorphism and the risk for suffering fibromyalgia. A total of 1531 fibromyalgia patients from 15 studies were included in this meta-analysis. It revealed an association between fibromyalgia and the Met/Met + Val/Met genotypes in the whole sample of patients. The Met allele of the COMT gene was associated with a higher impact of fibromyalgia compared to Val allele. Considering all this information, we have proceeded to edit the Discussion section trying to integrate our results with previous evidence providing similar findings (pages 21-22, lines 457-470).

Point 18 - The authors have changed wording to ‘patients/individuals with fibromyalgia’ and avoided ‘subjects’ for the most part. Yet, I would recommend them to do another check as some old terms are remaining (e.g., line 103, 354 or 421) and also the title still uses the old style. Also, talking about the title: I am not sure ‘fatigue perception’ is an appropriate term in this case.

The same with ‘catastrophism’ which is still used in the manuscript.

Response: Thank you for the reviewer’s suggestion. We have removed the term subjects in the entire manuscript. Likewise, we have changed the title and we have replaced the term fibromyalgia subjects for the term patients with fibromyalgia. In addition, also in the title, we have replaced the concept of fatigue perception by the term self-reported fatigue, as it has been used in the rest of the text. Finally, we have edited the manuscript to unify the use of the term catastrophism. We have used the term pain catastrophizing in the entire manuscript to refer this variable. 

Tables - The authors did a good job updating the result sections and corresponding tables.

Point 26 – It is still unclear how the Hardy-Weinberg equilibrium was tested. What was the expected distribution? And also, some of this discussion in the response letter would merit inclusion in the manuscript.

Response: We have included additional information on the purpose of the Hardy-Weinberg analysis in the Method section (pages 8-9, lines 195-202). In brief, Hardy-Weinberg method is a control analysis that is frequently conducted to analyse the distribution of genotypes within a given sample. Investigations focused on the study of associations between the presence of a disease and different genetic factors should met this equilibrium for the sample of healthy participants. It can also be done in the patients’ sample in order to explore if a specific gene could be associated with the development of a given disease. It would be an interesting question to explore in further investigations. In the current study, our aim was rather to explore the relationship between COMT gene and different symptoms characterizing this disease. For this reason, although following the reviewer’s suggestions we have added additional information about the purpose of this control analysis, we have not discussed about the distribution of genotypes in fibromyalgia since it adjusts to Hardy-Weinberg equilibrium. 

Point 37 – Wonderful. This would make an excellent contribution to the manuscript, so I would suggest including it.

Response: Following the reviewer's recommendations we have edited the manuscript including this information. We have added a paragraph in the Method section (see pages 9, lines 211-216), as well as the data corresponding to the correlation analysis (see Results section; page 14, lines 299-308). 

Point 41 – This is a nice discussion that also deserves to be added to the manuscript. Because no matter how you look at it, the one (and only) variable that is not elevated in the patients compared to controls does act as a moderator variable (even though analysis strategies are different). Discussion on why this would be, or at least acknowledging this, is very relevant.

Response: Following the recommendations given by the reviewer, we have added in the Discussion section, this information. This new information can be consulted on pages 22, lines 471-482.

General point - One of my struggles was with the use of terms such as genotype, (Methionine) alleles, polymorphisms, heterozygotes carriers, Val158Met, etc of which some seem to be synonyms, but others mean slightly different things.

Many things have been clarified in the responses as well as in the manuscript, for which I am grateful, but it would greatly improve the comprehensiveness of the manuscript if there was some consistency in referring to some of these jargon terms, especially in a journal like this which is read by a broad audience.

Response: Thanks for the reviewer's suggestion. We have revised the entire manuscript in order to clarify some of these terms. As the reviewer points out, PLOS ONE is a journal with a wide audience, but that fact should not undermine the use of appropriate terms, even if they are technical, to maintain certain of scientific quality. For that reason, we have maintained some these terms in the manuscript. 

Among other things, we have included the word polymorphism after the term Val158Met (e.g., see page 3, line 62 or page 5, line 97). Also, we have clarified that the Met alleles involve both the carriers of the Met/Met and Met/Val genotypes (e.g., page 3, line 68 or page 22, line 466). Finally, we have added the term gene after COMT in different parts of the manuscript (e.g., page 5 line 110 and 111).

Minor points

*Abstract line 33 “influences of COMT genotypes and group of participants” – does the latter refer to comparing patients with controls? Please rephrase.

Response: Thank you for your helpful comment. We have changed this sentence to reflect that the analysis shows the comparison between healthy participants and patients with fibromyalgia (see page 2, line 33-34).

*Page 6, lines 138-142 – This is not the place for this information, this should rather be placed at the description of the sample (around Table 2 e.g.)

Response: Thanks for your suggestion. As the reviewer indicates, we have moved the age analysis to the results section (page 11, line 251-254).

*Page 12, lines 265-266 – Please include the comparison Met/Met and Val/Val

Response: Following the reviewer's recommendations, we have included the comparison between homozygous subjects on page 13, lines 283-284.

*Page 12, line 268 – Please avoid qualifying words such as a ‘strong’ trend.

Response: Thank you for your comment. We have revised the manuscript and we have removed these qualifying words.

*Page 12 – It is unclear whether there were main effects of COMT gene (across groups).

Response: As you can see on the page 13, lines 290-291, there is information about the main effects of COMT genotype.

---

## [Decision Letter · Decision Letter 2]

11 Mar 2021

PONE-D-20-07498R2

Fear of pain moderates the relationship between self-reported fatigue and methionine allele of catechol-O-methyltransferase gene in patients with fibromyalgia

PLOS ONE

Dear Dr. Ferrera,

Thank you for submitting your manuscript to PLOS ONE. After careful consideration, we feel that it has merit but does not fully meet PLOS ONE’s publication criteria as it currently stands. Therefore, we invite you to submit a revised version of the manuscript that addresses the points raised during the review process.

The manuscript needs English language editing.

We look forward to receiving your revised manuscript.

Kind regards,

Mahmoud Abu-Shakra, MD

Academic Editor

PLOS ONE

Journal Requirements:

Reviewers' comments:

Reviewer's Responses to Questions

**Comments to the Author**

1. If the authors have adequately addressed your comments raised in a previous round of review and you feel that this manuscript is now acceptable for publication, you may indicate that here to bypass the “Comments to the Author” section, enter your conflict of interest statement in the “Confidential to Editor” section, and submit your "Accept" recommendation.

Reviewer #2: All comments have been addressed

Reviewer #3: All comments have been addressed

2. Is the manuscript technically sound, and do the data support the conclusions?

Reviewer #2: Yes

Reviewer #3: Yes

3. Has the statistical analysis been performed appropriately and rigorously? 

Reviewer #2: Yes

Reviewer #3: Yes

4. Have the authors made all data underlying the findings in their manuscript fully available?

Reviewer #2: Yes

Reviewer #3: Yes

5. Is the manuscript presented in an intelligible fashion and written in standard English?

Reviewer #2: No

Reviewer #3: Yes

6. Review Comments to the Author

Reviewer #2: The authors did an excellent job clarifying the manuscripts – with respect to patient-centered language, use of technical terms, the use of certain analyses (e.g., the grouping strategy), and discussion of important consideration. I support the publication of the manuscripts, although I would recommend a last language check. Especially in the revised parts, there are several grammatical errors. A few examples (not exhaustive) include:

- line 75-76 “because Met allele it is responsible for [..]”

- line 79-80 “due to a less release”

- line 301-302 “the higher was the score [..], the higher the score of [..]”

Reviewer #3: Accept as is all issues had been addressed I believe that the paper in its current form fits the scope of the journal

7. PLOS authors have the option to publish the peer review history of their article (what does this mean?). If published, this will include your full peer review and any attached files.

Reviewer #2: No

Reviewer #3: No

---

## [Author Response · Author response to Decision Letter 2]

3 Apr 2021

Author's response to reviewers

ID PONE-D-20-07498R2

"Fear of pain moderates the relationship between self-reported fatigue and methionine allele of catechol-O-methyltransferase gene in patients with fibromyalgia"

Authors:

David Ferrera (david.ferrera@urjc.es)

Francisco Mercado

Irene Peláez

David Martínez-Íñigo

Roberto Fernandes-Magalhaes

Paloma Barjola

Carmen Écija

Gema Díaz-Gil

Francisco Gómez-Esquer

Version: 4 Date: April 1, 2021

Dr. Mahmoud Abu-Shakra, MD

Academic Editor

PLOS ONE

Dear Dr. Abu-Shakra,

Thank you very much for your e-mail and comments submitted by the reviewers. Following your and the reviewer #2 suggestions, we have proceeded to carry out a professional English proofreading of the entire manuscript. We believe that our manuscript has been considerably improved as a result of these revisions and hope that this new version is acceptable for publication in PLOS ONE. We would like to thank you again for your consideration of our research and inviting us to submit the revised manuscript.

Best regards,

David Ferrera

---

## [Editor Report · Decision Letter 3]

12 Apr 2021

Fear of pain moderates the relationship between self-reported fatigue and methionine allele of catechol-O-methyltransferase gene in patients with fibromyalgia

PONE-D-20-07498R3

Dear Dr.Ferrera

We’re pleased to inform you that your manuscript has been judged scientifically suitable for publication and will be formally accepted for publication once it meets all outstanding technical requirements.

Kind regards,

Mahmoud Abu-Shakra, MD

Academic Editor

PLOS ONE
---

## [Editor Report · Acceptance letter]

14 Apr 2021

PONE-D-20-07498R3 

Fear of pain moderates the relationship between self-reported fatigue and methionine allele of catechol-O-methyltransferase gene in patients with fibromyalgia 

Dear Dr. Ferrera:

I'm pleased to inform you that your manuscript has been deemed suitable for publication in PLOS ONE. Congratulations! Your manuscript is now with our production department. 

Kind regards, 

on behalf of

Dr. Mahmoud Abu-Shakra 

Academic Editor

PLOS ONE